Fernández-Torras *et al. Genome Medicine*    (2019) 11:17

# Encircling the regions of the pharmacogenomic landscape that determine drug response

Adrià Fernández-Torras[1], Miquel Duran-Frigola[1*] and Patrick Aloy[1,2*]

## Abstract

**Background:** The integration of large-scale drug sensitivity screens and genome-wide experiments is changing the field of pharmacogenomics, revealing molecular determinants of drug response without the need for previous knowledge about drug action. In particular, transcriptional signatures of drug sensitivity may guide drug repositioning, prioritize drug combinations, and point to new therapeutic biomarkers. However, the inherent complexity of transcriptional signatures, with thousands of differentially expressed genes, makes them hard to interpret, thus giving poor mechanistic insights and hampering translation to clinics.

**Methods:** To simplify drug signatures, we have developed a network-based methodology to identify functionally coherent gene modules. Our strategy starts with the calculation of drug-gene correlations and is followed by a pathway-oriented filtering and a network-diffusion analysis across the interactome.

**Results:** We apply our approach to 189 drugs tested in 671 cancer cell lines and observe a connection between gene expression levels of the modules and mechanisms of action of the drugs. Further, we characterize multiple aspects of the modules, including their functional categories, tissue-specificity, and prevalence in clinics. Finally, we prove the predictive capability of the modules and demonstrate how they can be used as gene sets in conventional enrichment analyses.

**Conclusions:** Network biology strategies like module detection are able to digest the outcome of large-scale pharmacogenomic initiatives, thereby contributing to their interpretability and improving the characterization of the drugs screened.

## Background

Gene expression profiling has become a mainstay approach to characterize cell properties and status, unveiling links between gene activities and disease phenotypes. Early efforts were channeled into discovering transcriptional signatures that are specific to a disease state. This work involved the comparison of a relatively small number of diseased and healthy samples [1]. Although providing a rich account of disease biology, these studies have failed to yield better drug therapies, as causality and response to drug perturbations cannot be inferred

directly from two-state (diseased vs. healthy) differential gene expression analysis [2, 3]. To address this issue, initiatives have flourished to profile the basal gene expression levels of hundreds of cell lines, together with their response to treatment over an array of drug molecules using a simple readout such as growth rate [4–7]. Provided that the panel of cell lines is large enough, this approach allows for a new type of gene expression analysis where basal expression levels are *correlated* to drug response phenotypes. A series of recent studies demonstrate the value of this strategy for target identification, biomarker discovery, and elucidation of mechanisms of action (MoA) and resistance [8–13].

The largest cell panels available today are derived from cancerous tissues, since a crucial step towards personalized cancer medicine is the identification of transcriptional signatures that can guide drug prescription.

\* Correspondence: miquel.duran@irbbarcelona.org;
patrick.aloy@irbbarcelona.org
[1]Joint IRB-BSC-CRG Program in Computational Biology, Institute for Research in Biomedicine (IRB Barcelona), The Barcelona Institute of Science and Technology, Barcelona, Catalonia, Spain
Full list of author information is available at the end of the article

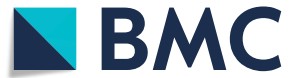

However, current signatures are composed of several hundred genes, thereby making them difficult to interpret, harmonize across platforms, and translate to clinical practice [14–16]. Recent assessment of sensitivity signatures for over 200 drugs [9] revealed that key genes include those involved in drug metabolism and transport. Intended therapeutic targets, though important, are detected in only a fraction of signatures, and cell line tissue of origin has been identified as a confounding factor throughout the signature detection procedure. In practice, the length of the signatures largely exceeds the number of sensitive cell lines available for each drug, which often yields inconsistent results between cell panels from different laboratories [14]. The current challenge is to filter and characterize transcriptional signatures so that they become robust, informative, and more homogeneous, while still retaining the complexity (hence the predictive power) of the original profiles [17].

Network biology offers means to integrate a large amount of omics data [18]. Most network biology capitalizes on the observation that genes whose function is altered in a particular phenotype tend to be co-expressed in common pathways and, therefore, co-localized in specific network regions [19]. Following this principle, it has been possible to convert genome-wide signatures to network signatures, or *modules*, that are less noisy and easier to interpret [20]. Raphael and co-workers, for instance, developed an algorithm to map cancer mutations on biological networks and identify "hot" regions that distinguish functional (driver) mutations from sporadic (passenger) ones [21]. Califano's group combined gene expression data with regulatory cellular networks to infer protein activity [22]. Overall, network-based methods come in many flavors and offer an effective framework to organize the results of omics experiments [23].

While many genes and proteins have enjoyed such a network-based annotation (being circumscribed within well-defined modules such as pathways and biological processes), drug molecules remain mostly uncharacterized in this regard. For a number of drugs, the mechanism of action is unclear [3] and off-targets are often discovered [24]. Recent publications of drug screens against cancer cell line panels, and the transcriptional signatures that can be derived from there, provide a broader view of drug activity and enable the full implementation of network biology techniques. Here we undertake the task of obtaining and annotating transcriptional modules related to 189 drugs. We show how these modules are able to capture meaningful aspects of drug biology, being robust to inherent biases caused by, for example, the cell's tissue of origin, and having a tight relationship to mechanisms of action and transportation events occurring at the membrane. Finally, we perform a series of functional enrichment analyses, which contribute to a better understanding of the molecular determinants of drug activity.

## Methods

### Data preparation and drug-gene correlations

We collected gene expression and drug response data from the GDSC resource (https://www.cancerrxgene.org). We first discarded those genes whose expression levels were low or stable across cell lines (Additional file 1: Figure S1A). To this end, we analyzed the distribution of basal expression of each gene in every CCL and filtered out those with an expression level below 4.4 (log2 units) across the panel (see Additional file 1: Figure S1B for a robustness analysis). Regarding drug response data, GDSC provides measurements of cell survival at a range of drug concentrations (area under the dose-response curve (AUC)). Since this measure is inversely proportional to drug sensitivity (i.e., the more sensitive the cell, the shorter its survival), we used the 1-AUC as a measure of potency. Thus, *positive* correlations denote drug sensitivity caused by gene overexpression while *negative* correlations indicate that sensitivity is associated with gene underexpression.

Recent studies report a confounding effect of certain tissues in the global analysis of drug-gene correlations [9]. In order to identify these potential biases in our dataset, we performed a principal component analysis (PCA) on the matrix of raw drug-gene correlations (Pearson's between 1-AUC and gene expression units). Then, we correlated the loadings of the first PC with gene expression values for each CCL. Finally, we filtered out CCLs belonging to tissues that were strongly correlated to the drug-gene correlation profiles (Additional file 1: Figure S2A). We removed leukemia, myeloma, lymphoma, neuroblastoma, small cell lung cancer (SCLC), and bone CCLs. In addition, we considered only drugs with sensitivity measurements available for at least 400 CCLs, as recommended by Rees et al. [9].

After this filtering process, we recalculated, for each drug-gene pair, the Pearson's correlation between basal gene expression and 1-AUC drug potencies across CCLs. We applied Fisher's z-transformation to the correlation coefficients in order to account for variation in the number of CCLs available for each drug [25]. Overall, we obtained positive and negative drug-gene correlations for 217 drugs and 15,944 genes across a total of 671 CCLs. Drug-gene correlations ($z_{cor}$) beyond $\pm 3.2$ were considered to be significant (Additional file 1: Figures S1C and S1D shows that this cutoff is a robust choice).

### Frequently correlated genes

For each gene, we counted the number of correlated drugs ($z_{cor}$ beyond $\pm 3.2$) and inspected the resulting

cumulative distribution (Additional file 1: Figure S3). Genes at the 5% end of the distribution were considered to be "frequently correlated genes" (FCGs). We found 869 positive and 799 negative FCGs, which were removed from further analyses. Finally, we performed enrichment analyses on those genes using the Gene Ontology database [26] and the DAVID toolbox (https://david.ncifcrf.gov/summary.jsp) (hypergeometric tests).

### Tissue-specific correlations

First, we split the CCL panel into sets of CCLs belonging to the same tissue. We then calculated drug-gene correlations ($z_{cor}$) separately for each of the 13 tissues represented in our dataset. In order to verify that measures of *positively* correlated genes (PCGs) and *negatively* correlated genes (NCGs) were consistent across tissues, we calculated the median $z_{cor}$ across tissues for each drug-PCG/NCG pair. In general, tissue-specific correlations had the same "direction" (i.e., same sign of $z_{cor}$) as the global correlation used throughout the study (Additional file 1: Figure S4A, left panel).

### Drug-target correlations

We obtained drug targets from the GDSC resource (disambiguating them with DrugBank [27], when necessary). We assigned at least one target to 202 of the 217 drugs. We focused on the $z_{cor}$ correlation of the targets to check whether target expression (positively) correlates with drug sensitivity. When more than one target was annotated per drug, we kept the maximum correlation. To validate the statistical significance of this measure, we randomly sampled genes (corresponding to the number of known targets per drugs; here again, we kept the maximum correlation). This process was repeated 1000 times for each drug. The mean and the standard deviation of this null distribution were used to derive a z-score, making results comparable between drugs.

### Drug module detection

After removing frequently correlated genes from the list of drug-gene correlations, we kept 182 [median; Q1: 84, Q3: 372] positively and 122 [median; Q1: 41, Q3: 337] negatively correlated genes (PCGs, NCGs) per drug. Further, we used correlation values ($z_{cor}$) to run a gene-set enrichment analysis (GSEA) [28] for each drug and identify the genes that participate in enriched Reactome pathways [29, 30]. We only considered Reactome pathways composed of at least 5 genes. Then, for each drug, we kept the significantly correlated genes found in any of the enriched pathways ($P$ value < 0.01). The resulting GSEA-filtered list of genes retained 100 [median; Q1: 49, Q3: 277] positive and 77 [median; Q1: 30, Q3: 221] negative correlations per drug. Then, taking the $z_{cor}$ values as input scores, we submitted the GSEA-filtered

list of genes to HotNet2 [31], using a high-confidence version of STRING [32] (confidence score > 700). We ran HotNet2 iteratively, keeping the largest module and removing its genes for the next iteration, until the modules had fewer than 5 genes or were not statistically significant ($p$ value > 0.05). To recall strong drug-gene correlations "proximal" to the drug modules (missed, most likely, by the incomplete coverage of Reactome), we used the DIAMOnD module-expansion algorithm [29]. We considered only genes that (i) were correlated to the drug response, (ii) were not present in any of the Reactome pathways, and (iii) were in the top 200 closest genes to the module, according to DIAMOnD (this cutoff was proposed by the authors of DIAMOnD based on orthogonal functional analyses). Hence, we obtained at least one positively correlated module for 175 of the drugs (48 genes [median; Q1: 23, Q3: 83]) and one negatively correlated module for 154 of the drugs (40 genes [median; Q1: 21, Q3: 78]). Robustness analysis of this procedure is found in Additional file 1: Figure S1D. A GMT list of the drug modules can be found in Additional file 2. The correlation values of the genes in the drug modules are available in Additional file 3.

### Distances between drug targets and modules

DIAMOnD [29] provides a list of genes sorted by their network-based proximity to the module. Accordingly, we retrieved from the STRING interactome the top closest 1450 genes (~ 10% of the largest connected component of the network) for every drug module. We then checked the ranking of drug targets in the resulting DIAMOnD lists, (conservatively) taking the median value when more than one target was available. To assess the proximity of drug targets to the modules, we measured distances to three different sets of random proteins. The first random set corresponded to the STRING proteome. For the second, we collected all genes defined as *Tclin* or *Tchem* in the Target Central Resource Database [33] (i.e., "druggable proteins"). Finally, the third random set included all pharmacologically active drug targets reported in DrugBank (https://www.drugbank.ca/).

### Distances between modules

We calculated distances between positively and negatively correlated modules separately using the network distance proposed by Menche et al. [34]. This distance measure is sensitive to the number of genes (size) included in the modules. To normalize this measure, we devised the following procedure. First, we grouped drug modules on the basis of their size. Then, for each module, we calculated the distribution of shortest distances from each gene to the most central one [35]. We used this distribution to sample random modules from the

network. When the distribution constraint could not be fully met, we used the DIAMOnD algorithm [29] to retrieve the remaining genes (50% of the genes at maximum). We repeated this process to obtain 10 random modules of each size. Next, we distributed the random modules into ranges (intervals) of 5 (i.e., from 10 to 14 genes, from 15 to 19, etc.; 50 random modules per interval). Then, for each pair size, we randomly retrieved 100 pairs of modules and calculated the network-based distance between them. The mean and standard deviation of the distances at each pair size were used to normalize the observed distances, correspondingly (z-score normalization) (we checked that 100 random pairs were sufficient to approximate the mean and standard deviation of the population). The more negative the network distance ($d_{net}$), the more proximal the modules are. We provide the network distances as an Additional file 4.

### Drug response prediction using drug modules

We performed drug response predictions in the GDSC dataset by using drug modules (only first PCMs and NCMs, to make results comparable between drugs). We devised a simple GSEA-like predictor in which CCLs were evaluated for their up-/downregulation of the modules, correspondingly. To this end, we first normalized the expression of each gene across the CCL panel (z-score). Then, for each drug, we ranked CCLs based on the GSEA enrichment scores (ES), taking drug modules as gene sets. To evaluate the ranking, we chose the top 25, 50, and 100 CCLs based on the *known* drug sensitivity profile. Performance was evaluated using the AUROC metric. Results were compared to those obtained with positively and negatively correlated genes (PCG, NCG) from the full signatures ($z_{cor}$ beyond ± 3.2).

To check whether modules derived from GDSC generalize to other CCL panels, we applied the same procedure to the Cancer Therapeutics Response Portal (CTRP) (https://ocg.cancer.gov/programs/ctd2/data-portal). As done with the GDSC panel, we removed all CCLs derived from neuroblastomas, hematopoietic, bone, and small cell lung cancer tissues, leaving a total of 636 CCLs, 397 in common with our GDSC panel (67 drugs in common). Drug response predictions for CTRP were performed as detailed above. We used the best ES among all modules associated with the drug. In addition, we did the analysis using CCLs exclusive to CTRP (i.e., not shared with the GDSC panel).

### Module enrichment in Hallmark gene sets

We downloaded the Hallmark gene set collection from the Molecular Signature Database (MSigDB) of the Broad Institute http://software.broadinstitute.org/gsea/index.jsp). We evaluated each gene set independently using a hypergeometric (Fisher's exact) test (first and second modules were merged, when applicable; the gene universe was that of GDSC). Enrichments can be found in the Additional file 5.

### Drug module enrichments in the TCGA cohort

We downloaded gene expression data (median z-scores) for 9788 patients and 31 cancer tissues from the Pan-Cancer Atlas available in the cBioPortal resource (http://www.cbioportal.org). "Presence" or "expression" of the module in each patient was evaluated using GSEA (*P* value < 0.001), ensuring that the direction (up/down) of the enrichment score corresponded to the "direction" of the module (PCM/NCM). For a complete list of enrichment results, please see Additional file 6 (results are organized by tumor type). Further, to identify associations between drug modules and cancer driver genes, we checked whether patients "expressing module of drug X" (*P* value < 0.001) were "harboring a mutation in driver gene Y" (Fisher's exact test). We considered 113 driver genes (obtained as described in [36], using the "known" flag) (Additional file 7).

### Characterization of drug modules

In order to characterize drug modules from different perspectives, we designed 21 features belonging to the following categories: (i) *General features* derived directly from the pharmacogenomics panel, (ii) *Network features* related to network measures such as topological properties, and (iii) *Biological features* encompassing a series of orthogonal analyses related to drug biology. For more information, please see Additional file 8 and its corresponding legend.

## Results and discussion

The Genomics of Drug Sensitivity in Cancer (GDSC) is the largest cancer cell line (CCL) panel available to date [8]. This dataset contains drug sensitivity data (growth-inhibition, GI) for 265 drugs screened against 1001 cell lines derived from 29 tissues, together with *basal* transcriptional profiles of the cells (among other omics data). Aware of the work by Rees et al. [9], we first looked for the dominant effect of certain tissues in determining associations between drug response and gene expression. We found that CCLs derived from neuroblastoma, hematopoietic, bone, and small cell lung cancers may confound global studies of drug-gene correlations due to their unspecific sensitivity to drugs (Additional file 1: Figure S2A). These tissues were excluded from further analyses. We also excluded genes whose expression levels were low or constant across the CCL panel and drugs tested against fewer than 400 CCLs (see the "Methods" section for details). As a result, we obtained a pharmacogenomic dataset composed of 217 drugs, 15,944 genes, and 671 CCLs.

Following the conventional strategy to analyze pharmacogenomic datasets, we calculated *independent* drug-gene associations simply by correlating the expression level of each gene to the potency of each drug (area over the growth-inhibition curve; 1-AUC) across the CCL panel. We used a *z*-transformed version of Pearson's , as recommended elsewhere [25]. Figure 1a shows the pair-wise distribution of the *z*-correlation ($z_{cor}$) measures between the 15,944 genes and the 217 drugs. We validated the correlations identified in the GDSC panel on an independent set by applying the same protocol to the Cancer Therapeutic Response Portal (CTRP) panel [9] (Additional file 1: Figure S4B). To identify the strongest drug-gene associations, we set a cutoff of ± 3.2 $z_{cor}$, based on an empirical null distribution obtained from randomized data (see Additional file 1: Figure S1C and the "Methods" section). Please note that this is a widely adopted procedure that is not designed to detect *single* drug-gene associations (which would require multiple testing correction) [37]. Instead,

and similar to signature identification in differential gene expression analysis, the goal is to identify sets of genes that are (mildly) correlated with drug response. For each drug, we obtained a median (Med) of 249 positively correlated genes [first quartile (Q1): 120, third quartile (Q3): 584], and Med of 173 negatively correlated genes [Q1: 59, Q3: 484] (Fig. 1b). Some drugs, like the BRAF inhibitor dabrafenib, or the EGFR inhibitor afatinib, had over 1500 positively and negatively correlated genes, while others, like the antiandrogen Bicalutamide or the p38 MAPK inhibitor Doramapimod, had hardly a dozen. We observed that the number of genes that correlate with drug response strongly depends on the drug class (Fig. 1c), EGFR and ERK-MAPK signaling inhibitors being the classes with the largest number of associated genes, and JNK/p38 signaling and chromatin histone acetylation inhibitors being those with the fewest correlations. This variation may be partially explained by the range of drug potency across the CCL panel, as it is "easier" to detect drug-gene correlations when the drug

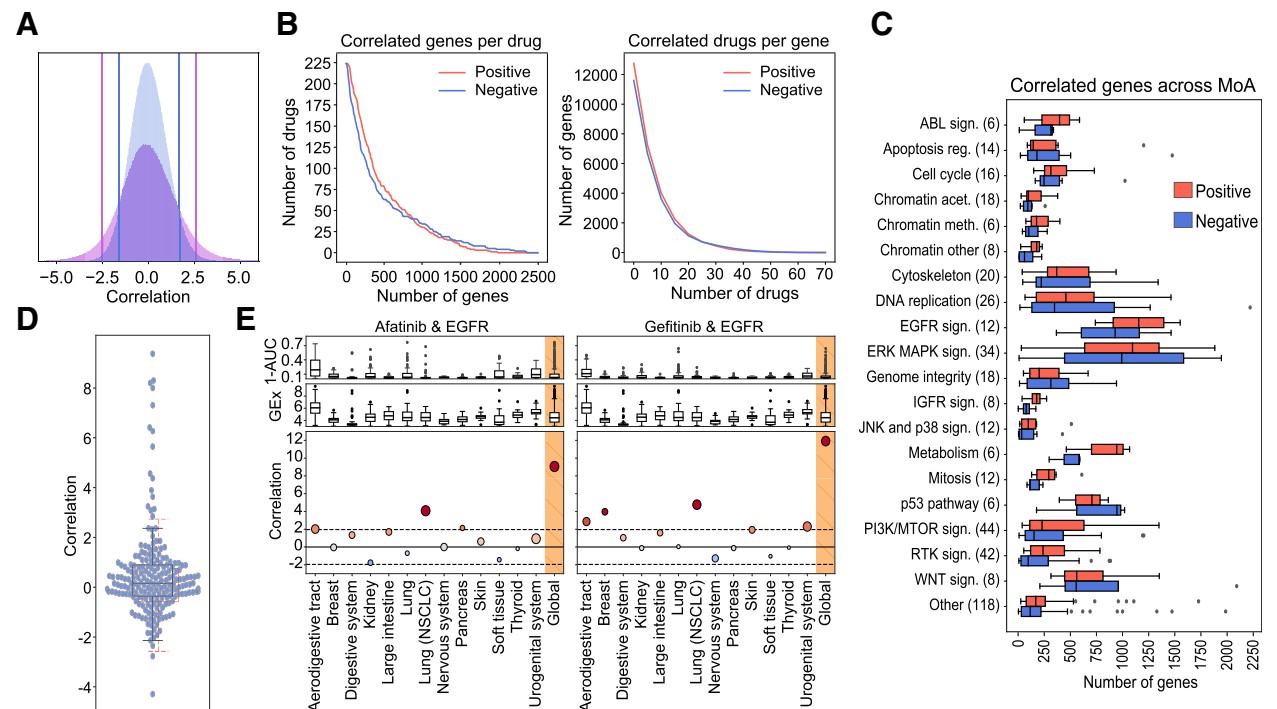

**Fig. 1** Analysis of drug-gene correlations. **a** Observed drug-gene correlation distribution (purple) and randomized drug-gene correlation distribution (blue) (random permutation of expression values). Vertical lines denote the percentiles 5 and 95 of each distribution. **b** The left panel shows the "number of correlated genes per drug", while the right panel shows the "number of correlated drugs per gene". In the left panel, one can read, for example, that there are about 25 drugs (*y*-axis) with at least 1250 correlated genes (*x*-axis). Likewise, in the right panel, one can read that about 4000 genes (*y*-axis) are correlated to at least 10 drugs (*x*-axis). **c** Number of positively (red) and negatively (blue) correlated genes across drug classes. **d** Positively correlated targets (see the "Methods" section for details on the *z*-score normalization procedure of this correlation measure). Each dot represents one drug-target correlation. A full account of drug-target annotations is provided in Additional file 8. The red boxplot shows the background (random) distribution. **e** Drug-gene correlations ($z_{cor}$) between afatinib/gefitinib and the epidermal growth factor receptor (EGFR) across tissues. In the upper plots, we show the drug sensitivity (1-AUC) across tissues. In the middle plots, we show basal gene expression of EGFR across tissues. Bottom plots show the Afatinib/Gefitinib-EGFR correlation. The rightmost values refer to the correlation when all tissues are considered (Global). Size of the bubbles is proportional to the number of CCLs in each tissue

has a wide sensitivity spectrum (Additional file 1: Figure S5).

Similarly, analysis of independent drug-gene correlations suggests that some genes are positively correlated to many drugs. For instance, we found 5% of the genes to be associated with more than 10% of the drugs (Fig. 1b and Additional file 1: Figure S3). The transcripts of these "frequent positively correlated genes" are enriched in membrane processes, specifically focal adhesion ($P$ value $< 5.2 \times 10^{-12}$) and extracellular matrix (ECM) organization ($P$ value $< 5 \times 10^{-16}$), including subunits of integrin, caveolin, and platelet-derived growth factors (PDGFs). These genes determine, among others, the activation of Src kinases [38–41]. Overall, ECM proteins are known to play an important role in tumor proliferation, invasion, and angiogenesis [42, 43] and are often involved in the upstream regulation of cancer pathways [44] such as PI3K/mTOR [38–40], MAPK [39], and Wnt signaling [45], and in cell cycle and cytoskeleton regulation [46]. It is thus not surprising that ECM genes determine drug response in a rather unspecific manner.

On the other hand, "frequent negatively correlated genes" are associated with small molecule metabolism (xenobiotic metabolic processes, $P$ value $< 3.2 \times 10^{-3}$). In this group, we found, among others, the cytochrome CYP2J2 and the GSTK1 and MGST glutathione transferases, which are highly expressed in cancers and known to confer drug resistance through their conjugating activity [47–50]. Following other studies that reported similar results [9], we checked for the presence of multidrug transporters (MDTs). Reassuringly, we found the efflux pump transporter ABCC3 and a total of 27 different solute carriers (SLCs) to be negatively correlated to the potency of many drugs. Of note, we also found the ABCA1 transporter and other 8 SLCs to be among the frequent positively correlated genes, thus emphasizing the key role of transporters and carriers in determining drug potency.

All of the above suggests that systematic analysis of independent drug-gene correlations is sufficient to highlight *unspecific* determinants of drug sensitivity and resistance (i.e., frequent positively and negatively correlated genes). However, while these determinants are recognized to play a crucial role, they do not inform targeted therapies, as they are usually unrelated to the mechanism of action of the drug. Thus, we assessed whether measuring drug-gene correlations would also be sufficient to elucidate drug targets, i.e., we tested whether the expression level of the target correlates with the potency of the drug. Since most drugs had more than one annotated target, to measure significance, we randomly sampled 1000 times an equal number of genes and derived an empirical $z$-score (see the "Methods" section). Figure 1d shows that the expression level of most drug targets did *not* correlate with drug response. In fact, only ~ 10% of the drugs had "positively correlated targets" ($z$-score $> 1.9$, $P$ value ~ 0.05). Remarkably, the 6 EGF pathway inhibitors in our dataset were among these drugs, as were 3 of the 4 IGF pathway and 3 of the 21 RTK pathway inhibitors. We noticed that the molecular targets for these pathways were usually cell surface receptors, e.g., EGFR, IGFR, ALK, ERBB2, MET, and PDGFRA. Overall, of the 20 drugs with positively correlated targets, 13 bind to cell surface receptors, showing a propensity of drug-gene correlations to capture membrane targets (odds ratio = 15.13, $P$ value = $1.9 \times 10^{-7}$). In Additional file 1: Figure S6, we show how this trend is driven mostly by the over-expression of the target on the cell surface.

The relatively small number of positively correlated targets illustrates how the analysis of expression levels alone is insufficient to reveal MoAs, especially when the drug target is located downstream of the cell surface receptors in a signaling pathway. Some authors have suggested that the tissue of origin of the cells might play a confounding role in defining drug response signatures. To address this notion, we repeated the calculation of Pearson's $z_{cor}$ correlations separately for each of the 13 tissues in our dataset. In general, the trends observed at the tissue level were consistent with the global trends, although tissue-specific correlations were milder due to low statistical power (i.e., few cell lines per tissue) (Additional file 1: Figure S4A, right panel). Accordingly, we confirmed that none of the tissues had a globally dominant effect on the measures of drug-gene correlations (Additional file 1: Figure S2B) and verified that certain tissue-specific associations were still captured by the analysis. For instance, going back to the targeting of EGFR (which was positively correlated with Afatinib and Gefinitib), we show in Fig. 1e that the "global" correlation can be partly attributed to non-small cell lung cancer (NSCLC) cells ($z_{cor} > 1.96$, $P$ value $< 0.05$). Indeed, afatinib and gefitinib have an approved indication for NSCLC. Both drugs correlate with EGFR also in the aerodigestive tract, an observation reported in an independent study dedicated to the discovery of drug-tissue/mutation associations (ACME) [7]. Moreover, and consistent with recent findings [51–54], gefitinib has a significant correlation to EGFR in breast cancers, whereas afatinib correlates with this target in pancreatic CCLs. Afatinib, in turn, is associated with ERBB2 in breast CCLs, as also confirmed by ACME analysis (Additional file 1: Figure S4C).

## From drug-gene correlations to drug modules

The previous analysis demonstrates that conventional drug-gene correlations do *not* directly identify drug targets and suggests that standard transcriptional drug

signatures contain unspecific and indirect correlations that may mislead mechanistic interpretation. Recent advances in network biology precisely tackle these problems, as they can (i) filter signatures to make them more functionally homogeneous and (ii) allow for the measurement of network distances so that genes proximal to the target can be captured and connected to it, even if the expression of the target itself is not statistically associated with the drug.

Hence, we set to mapping drug-gene correlations onto a large protein-protein interaction (PPI) network, retaining only genes that could be grouped in network *modules* (i.e., strongly interconnected regions of the network). In the "Methods" section, we explain in detail the module detection procedure. In brief, starting from drug-gene correlations (Fig. 2A), we first filtered out those genes whose expression was frequently (and unspecifically) correlated to the potency of many drugs (Additional file 1: Figure S3). This reduced the number of associations to 182 [median; Q1: 84, Q3: 372] positively and 122 [median; Q1: 41, Q3: 337] negatively correlated genes per drug, respectively. Next, in order to identify genes acting in coordination (i.e., participating in enriched Reactome pathways [29, 30]), we adapted the gene set enrichment analysis (GSEA) algorithm [28] to handle drug-gene correlations (instead of gene expression fold-changes) (Fig. 2B). The resulting GSEA-filtered list of genes kept 100 [median; Q1: 49, Q3: 277] positive and 77 [median; Q1: 30, Q3: 221] negative correlations per drug. After this filtering, we submitted this list to HotNet2 [31], a module detection algorithm that was originally developed for the identification of recurrently mutated subnetworks in cancer patients (Fig. 2C; Additional file 1: Figure S7 shows the importance of the Reactome-based filtering previous to HotNet2). As a reference network (interactome) for HotNet2, we chose a high-confidence version of STRING [32], composed of 14,725 proteins and 300,686 interactions. HotNet2 further filtered the list of genes correlated to each drug, keeping only those that were part of the same network neighborhood. Finally, we used the DIAMOnD module expansion algorithm [29] to recover strong drug-gene correlations that had been discarded along the process. Although this step made a relatively minor contribution to the composition of the modules (less than 5% of the genes; Additional file 1: Figure S8), we did not want to lose any strong association caused by the limited coverage of the Reactome database (Fig. 2D).

Our pipeline yielded at least one "positively correlated module" (PCM) for 175 of the 217 drugs (48 genes [median; Q1: 23, Q3: 83]). Similarly, we obtained "negatively correlated modules" (NCMs) for 154 of the drugs (40 genes [median; Q1:21, Q3:78]). Thus, compared to the original signatures, drug modules are considerably smaller (80% reduction) (Fig. 3a) and are commensurate with manually annotated pathways in popular databases (Additional file 1: Figure S9). For roughly two thirds of the drugs, we obtained only one PCM and one NCM. For the remaining drugs, a second (usually smaller) module was also identified (Additional file 1: Figure S10A). The complete list of drug modules can be found in Additional file 2. Pair-wise drug-gene correlations of the modules are listed as Additional file 3. Additionally, the code of the module-detection pipeline is available at: https://github.com/sbnb-lab-irb-barcelona/GDSC-drug-modules.

## Drug modules are tightly related to mechanisms of action

To assess the mechanistic relevance of drug modules, we measured their distance to the corresponding drug targets, i.e., we formulated the hypothesis that drug targets should be "proximal" to dysregulated network regions. To this end, we used the DIAMOnD algorithm again [29], this time to retrieve, for each drug, a list of genes ranked by their proximity to the corresponding drug module(s) (see the "Methods" section). Figure 3b shows that drug targets are remarkably up-ranked in these lists, making them closer to the drug modules than any other set of random proteins, including druggable genes and pharmacological receptors [33], which usually have prominent positions in the PPI network due to the abundant knowledge available for them. In 82% of the PCMs, the corresponding targets were among proximal proteins (top decile), which means a dramatic increase in mechanistic interpretability compared to the 12.25% of drugs that could be linked to their targets via conventional analysis of drug-gene correlations.

A unique feature of drug modules is that network-based distances can be natively measured between them [34]. We computed the distance between drug modules pair-wise (Additional file 4) and grouped them by drug class (Fig. 3c) (see the "Methods" section and alternative statistical treatments in Additional file 1: Figure S11). The diagonal of Fig. 3c clearly indicates that drugs belonging to the same category tend to have "proximal" modules (some of them in a highly specific manner, like in the case of ERK-MAPK signaling cascade inhibitors). Most interestingly, we could observe proximities between modules belonging to different drug classes. For instance, modules of drugs targeting RTK signaling were "located" near to those of drugs affecting genome integrity, in good agreement with recently reported cross-talk between these two processes [55, 56]. Likewise, and as proposed by some studies [57–59], IGFR-related drugs were "proximal" to drugs affecting cell replication events such as mitosis, cell cycle, and DNA replication.

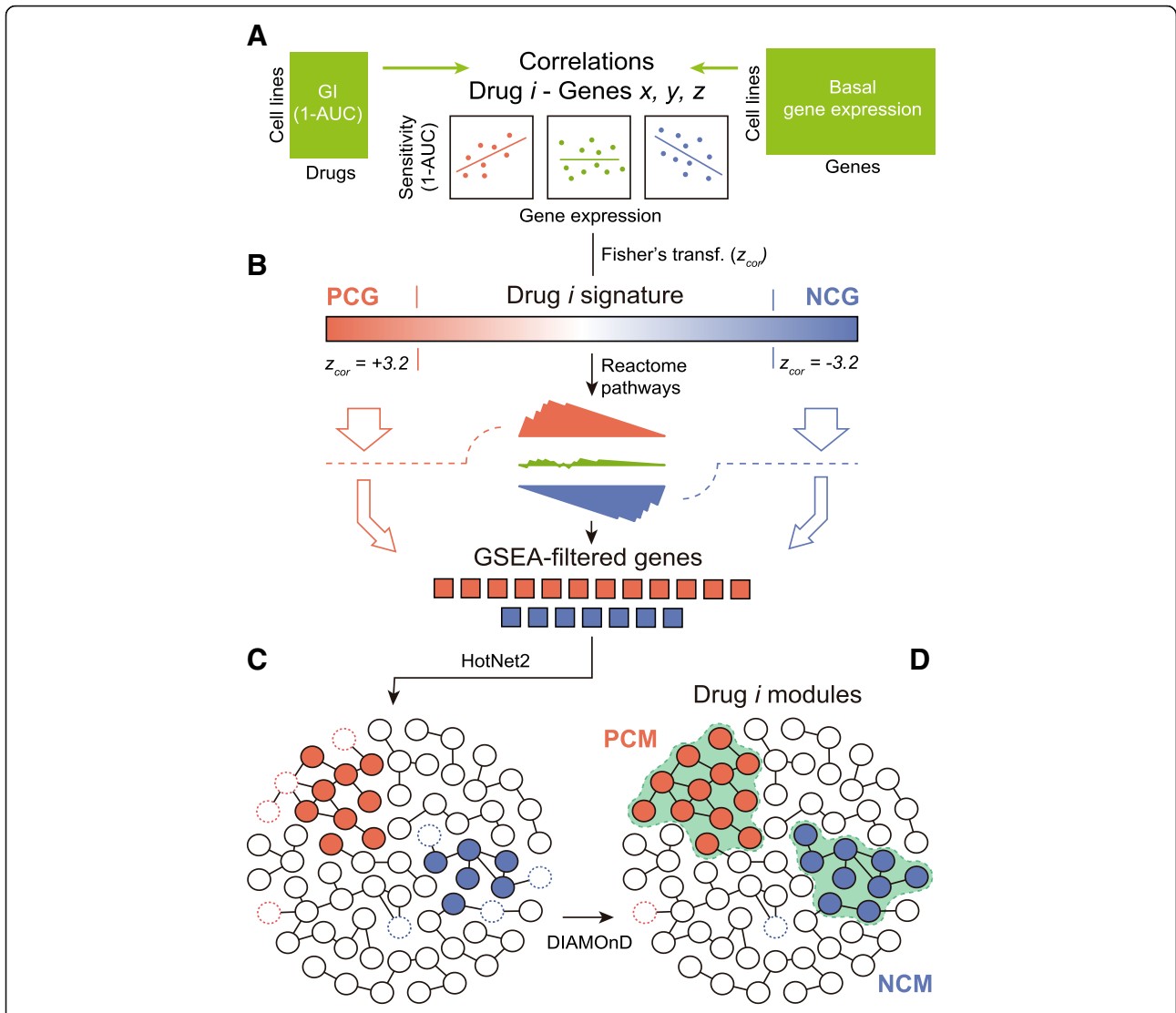

**Fig. 2** Methodological pipeline to identify drug modules. (A) The process of obtaining modules starts with the calculation of z-transformed Pearson's correlation ($z_{cor}$) between gene expression and drug sensitivity data for each drug-gene pair. Correlations ($z_{cor}$) beyond ± 3.2 are considered to be significant. (B) We then run a gene-set enrichment analysis (GSEA) for each drug in order to identify genes that participate in enriched Reactome pathways. (C) GSEA-filtered genes are submitted to HotNet2 on the STRING interactome in order to identify drug modules. (D) Finally, modules are expanded (when possible) using the DIAMOnD algorithm to recall the few correlated genes that might have been excluded in step C as a result of the limited coverage of the Reactome database. This final step has a minor impact on the composition of the module

## Drug modules retain the ability to predict drug response

We have shown that drug modules are related to the MoA of the drug, but the question remains as to the extent to which they retain the predictive capabilities of the full transcriptional profiles/signatures. In the CCL setting, gene expression profiles are valuable predictors of drug response [5, 11, 60] and crucially contribute to state-of-the-art pharmacogenomic models. To test whether our (much smaller) drug modules retained predictive power, we devised a simple drug sensitivity predictor based on the GSEA score (see the "Methods" section). In brief, given a drug, we tested whether cell

lines *sensitive* to a certain drug were enriched in the corresponding drug modules. We expect genes in PCMs to be *over*-expressed in sensitive cell lines and those in NCMs to be *under*-expressed. Analogously, we took the positively and negatively correlated genes from the full drug-gene associations (signatures) and also performed a GSEA-based prediction. To nominate a cell "sensitive" to a certain drug, we ranked CCLs by their sensitivity and kept the top $n$ CCLs, $n$ being 25, 50, or 100, based on the distribution of sensitive cell lines provided by the authors of the GDSC (Additional file 1: Figure S12A). This simple binarization is, in practice, proportional to

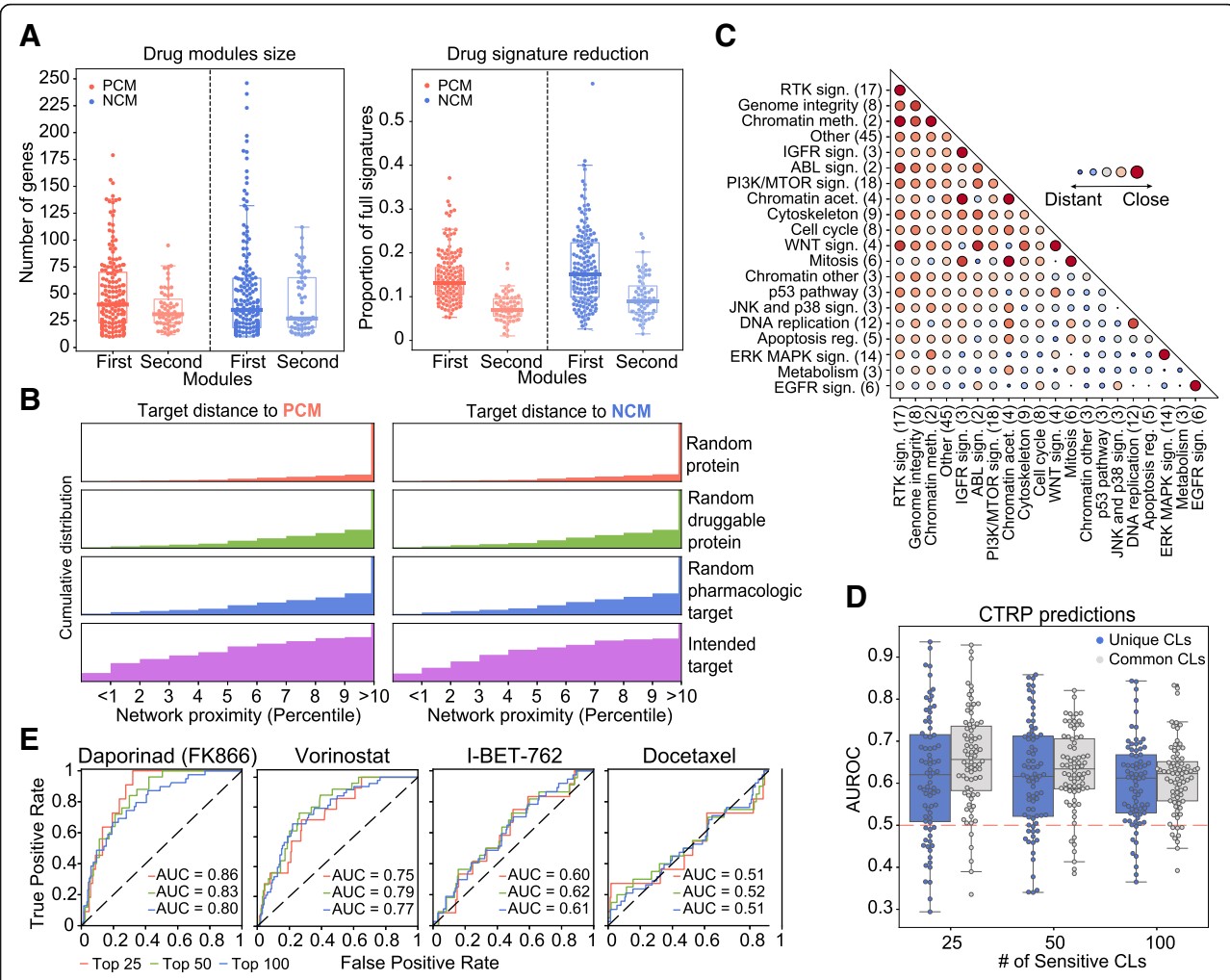

**Fig. 3** Global drug module analysis. **a** Number of genes in positively and negatively correlated modules (PCMs and NCMs) (left). Proportion of genes in the modules with respect to PCGs and NCGs (i.e., full signature). **b** Distance between drug targets and PCMs/NCMs (purple cumulative distribution). Results are compared to random proteins from the STRING interactome (red), proteins sampled from the "druggable proteome" (Target Central Resource Database) (green), and proteins sampled from the pharmacological targets in DrugBank (blue). **c** Network-based distances between drug classes. The bigger the bubble, the closer the distance between drug classes. Drug classes are sorted by specificity in their proximity measures. Please note "distant" values in the diagonal are possible due to differences in drug modules belonging to the same class. The network-based distance calculation is detailed in the "Methods" section. **d** Predictive performances (AUROC) of the drug modules evaluated in the CTRP panel (top 25, 50, and 100 sensitive CCLs). Blue distributions correspond to results using unique CCLs (i.e., not shared with the GDSC panel). **e** Illustrative ROC curves for Daporinad (FMK866), Vorinostat, I-BET-762, and Docetaxel

more sophisticated "sensitive/resistant" categorizations such as the waterfall analysis [14], and it yields prediction performance metrics comparable between drugs.

Additional file 1: Figure S13 suggests that, when applied to the GDSC, drug module enrichment analysis can classify sensitive cell lines with high accuracy, especially for the top 25 sensitive cell lines (area under the ROC curve (AUROC) 0.77), which is a notable achievement considering that drug modules are 80% smaller than the original signatures. To assess the applicability of our modules outside the GDSC dataset, we performed an external validation with the CTRP panel of cell lines. About 37% of our drugs were also tested in this panel.

In CTRP, drug sensitivity is measured independently of GDSC, which poses an additional challenge for prediction as a result of experimental inconsistencies [14]. Of the CCLs, 397 are shared between GDSC and CTRP, and gene expression data are also measured independently. We performed the GSEA-based sensitivity prediction for *all* CTRP CCLs. Figure 3d and e show the distribution of prediction performances for the 70 drugs, and illustrative ROC curves corresponding to four drugs (namely Daporinad, Vorinostat, I-BET-762 and Docetaxel), respectively. We found that, when focusing on the top 25 sensitive CCLs, over a quarter of the drugs had AUROC > 0.7, including Daporinad. Acceptable

(AUROC > 0.6) predictions were achieved for half the cases (e.g., Vorinostat and I-BET-762), which is a comparable result to recent attempts to translate sensitivity predictors between different CCL panels [61]. For the remaining drugs, predictive performance did not differ to random expectation (AUROC < 0.6) (e.g., docetaxel). Notably, performance declined only slightly when considering CCLs that were *exclusive* to the CTRP panel (i.e., not part of the GDSC dataset) (Fig. 3d, blue boxes). The figure was comparable, if not better, to that obtained using full signatures (PCGs and NCGs) (Additional file 1: Figure S13, gray boxes). These observations support previous recommendations to pre-filter pharmacogenomic data based on prior knowledge [62] (Additional file 1: Figure S14).

## Module-based characterization of drugs

Since drug modules are highly connected in biological networks, they are expected to be (at least to some extent) functionally coherent and easier to interpret. Accordingly, we tested the enrichment of drug modules in a collection of high-order biological processes (the Hallmark gene sets) available from the Molecular Signatures Database (MSigDB) [63]. Additional file 1: Figure S15A shows that the number of enriched Hallmark gene sets depends upon the MoA of the drug. The results of the enrichment analysis are given in Additional file 5 and as an interactive exploration tool based on the CLEAN methodology (Additional file 9; https://figshare.com/s/932dd94520d4a60f076d) [64]. We chose three drug classes to illustrate how to read these results, namely drugs targeting mitosis, RTK signaling inhibitors, and ERK-MAPK signaling inhibitors (Fig. 4a).

Drugs targeting mitosis have modules enriched in cell cycle and replication processes (Fig. 4a, top). Specifically, genes related to the Myc transcription factor are over-represented in three of the drug modules (NPK76-II-72-1, GSK1070916, and MPS-1-IN-1). The modules of these drugs have a rather distinct composition, NPK76-II-72-1 having the largest coverage of Myc-related genes and being, together with MPS-1-IN-1, related to both Myc1 and Myc2 processes. In Additional file 1: Figure S15B, we show how, for these two drugs, cell line sensitivity is dependent on Myc expression levels.

In contrast to mitosis inhibitors, drugs targeting the RTK pathway are enriched in biological processes outside the nucleus (Fig. 4a, middle), among these hypoxia and the epithelial-mesenchymal transition (EMT). Both mechanisms are known to be associated with tyrosine kinases [65, 66]. Interestingly, a subgroup of RTK inhibitors (namely ACC220, CEP-701, NVP-BHG712, and MP470) is characteristically associated with the PI3K-AKT-mTOR signaling cascade. With the exception

of NVP-BHG712, these inhibitors have the tyrosine kinase FLT3 as a common target [67, 68]. Deeper inspection of FLT3 inhibitors reveals module proximities to certain PI3K inhibitors (e.g., GDC0941), and the PI3K-AKT-mTOR pathway is enriched in ERBB2 inhibitors as well (Additional files 4 and 5).

As for ERK-MAPK pathway inhibitors, we observed a total of 17 enriched Hallmarks, making this class of drugs the one with most variability in terms of enrichment signal of the modules (Fig. 4a, bottom; Additional file 1: Figure S15A). However, while some processes like apoptosis are detected in most of the drugs in this category, others are target-specific. Oxidative phosphorylation (OXPHOS), for example, is represented in 3 of the 4 BRAF inhibitors. It is known that, while BRAF inhibitors boost OXPHOS (leading to oncogene-induced senescence), activation of glycolytic metabolism followed by OXPHOS inactivation yields drug resistance [69, 70]. Similarly, VX11e (the only drug in our dataset targeting ERK2) shows a distinctive enrichment in Myc-regulated proteins, while FMK (the only drug targeting the Ribosomal S6 kinase) is enriched in p53 signaling pathway and inflammatory response processes. All these observations are consistent with previous studies [71–74], and Additional file 1: Figure S15C demonstrates that the variability observed between drugs in this class is driven mostly by differences in the sensitivity profiles of the drugs.

Overall, the enrichment signal (i.e., the functional coherence) of drug modules is substantially higher than that of full signatures (PCGs and NCGs) (Fig. 4b,c). This facilitates, in principle, the mechanistic interpretation of drug-gene correlation results (Additional file 1: Figure S15D). We show an illustrative module (CEP-701) in Fig. 4d.

We next examined whether our results could be extended beyond CCL panels. We found that drug modules are indeed identified (GSEA *P* value < 0.001) in the majority of patients in the TCGA clinical cohort (Additional file 1: Figure S15E; see the "Methods" section for details). Closer inspection by TCGA tumor type further supports the clinical relevance of our results (Additional file 6). For example, drugs affecting MAPK signaling (specifically, BRAF inhibitors, e.g., dabrafenib) have a tendency to "occur" in skin cutaneous melanomas (SKCM), as expected (Fig. 4e, blue). Of note, one PPAR inhibitor (FH535) was also found enriched in a high number of SKCM patients, in good agreement with work by others proposing the use PPAR inhibitors to treat skin cancer [75, 76]. Similarly, we observed an abundance of EGFR inhibitor modules among pancreatic cancers (PAAD) (Fig. 4e, green), in line with the known crucial role of EGFR in pancreatic tumorigenesis [77, 78]. As for gliobastomas (GBMs) (Fig. 4e, purple), we found two GSK3 inhibitors (CHIR-99021 and SB216763) and one TNKS inhibitor (XAV939), all of them targeting

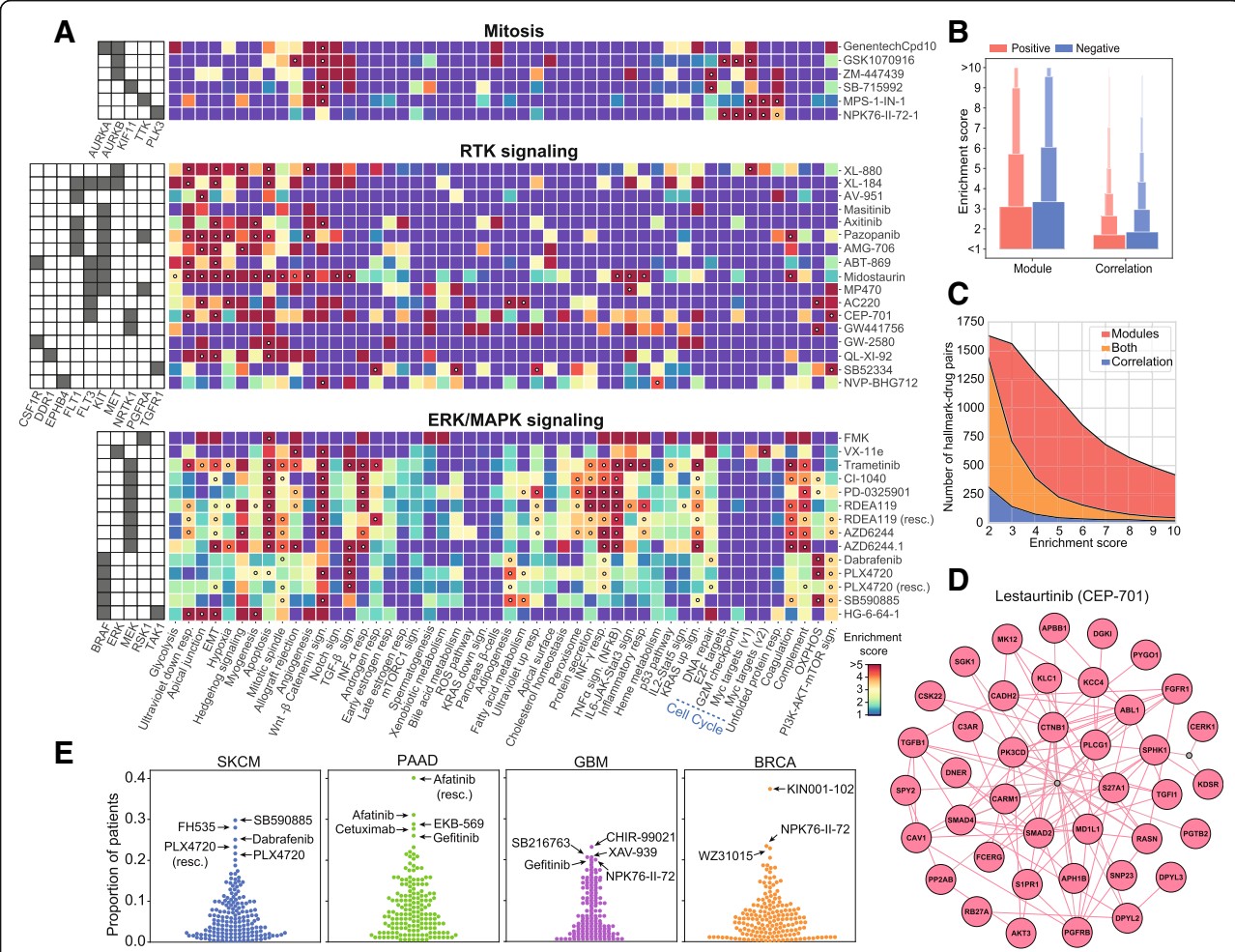

**Fig. 4** Drug module characterization. **a** Drug module enrichment analysis based on the Hallmark gene set (odds ratios in color, *p* values < 0.05 are marked with a white dot). For simplicity, three drug classes are shown: drugs affecting mitosis, RTK signaling, and ERK/MAPK signaling. **b** Distribution of the enrichment scores in the Hallmark collection gene sets. Overall, higher enrichment scores are obtained using modules than using full signatures (PCGs and NCGs) (the gene universe used here is that of Reactome). **c** Similarly, number of Hallmark-drug pairs at different enrichment scores. We show the pairs found only with the modules (PCMs and NCMs, red), only with correlations (PCGs and NCGs, blue), or in both (orange). **d** A view of Lestaurtinib (CEP-701) module. For illustrative purposes, two out-of-the-module (non-correlated) proteins are shown (gray), one being very central and one being peripheral but acting as a "bridging" node. **e** TCGA enrichment analysis of PCMs in four types of cancer: SKCM (Cutaneous Melanoma), PAAD (Pancreatic Adenocarcinoma), GBM (Glioblastoma Multiforme), and BRCA (Breast Carcinoma)

WNT signaling, which is a potential mechanism against this tumor type [79]. We also found one EGFR inhibitor (Gefitinib) and the PLK inhibitor NPK76-II-72-1 mentioned above in the context of Myc enrichment analysis. Both mechanisms have shown promise in EGFR- and Myc-activated gliomas, respectively [80, 81]. Finally, we encountered a more heterogeneous pattern in breast cancer patients (BRCA) (Fig. 4e, orange), including mechanisms supported by the literature, such as AKT, IRAK1, and PLK3 inhibition [82–84].

Beyond the tumor-type level, we looked for modules that were significantly enriched (odds ratio > 2, *P* value < 0.001) in patients harboring specific driver mutations (see the "Methods" section). A full account of this enrichment

analysis is given in Additional file 7. We found, for instance, that modules of drugs targeting ERK/MAPK signaling are related to patients with mutations in HRAS and BRAF [85, 86] and that, in turn, BRAF is (together with KRAS) frequently mutated in patients "expressing" modules of EGFR signaling inhibitors [87]. Taken together, and although TCGA treatment response data is too scarce to allow for prediction assessment [88], these results indicate that the drug modules identified in CCLs hold promise for translation to clinical practice.

## Conclusions

Two limitations of large-scale pharmacogenomic studies are the difficulty to reproduce results across screening

platforms and the eventual translation to clinics, as it remains unclear whether immortalized cells are able to model patient samples [89]. Another important limitation is the overwhelming number of drug-gene correlations that can be derived from these experiments, yielding signatures of drug sensitivity that are almost impossible to interpret. We have shown, for example, that (i) the number of correlated genes is highly variable across drugs, (ii) some genes are unspecifically correlated to many drugs, and (iii) not all drug-gene pairs are equally correlated in every tissue. We propose that converting transcriptional signatures to network modules may simplify the analysis, since network modules are smaller, more robust, and functionally coherent. We have validated this strategy by proving that drug response modules, which are enriched in biological processes of pharmacological relevance and exhibit comparable predictive power to the full signatures, are tightly related to the MoA. Further, we have characterized the modules extensively (Additional file 8 and e.g., Additional file 1: Figure S16) and confirmed their occurrence in the TCGA clinical cohort (Additional file 6 and Additional file 10).

However, our approach does have some of the limitations of ordinary transcriptomic analyses. Expression levels of mRNA do not perfectly match protein abundance, nor are they able to capture post-translational modifications such as phosphorylation events, which are key to some of the pathways studied here. Moreover, wide dynamic ranges in gene expression and drug sensitivity data are necessary for drug-gene correlations to be captured, thus requiring, in practice, considerably large panels of CCLs, which limits the throughput of the technique to a few hundred drugs. In particular, one cannot precisely measure correlations within poorly represented tissues, which in turn makes it difficult to disentangle tissue-specific transcriptional traits that may be irrelevant to drug response. Our module-based approach partially corrects for this confounding factor, although the integration of other CCL omics data (such as mutations, copy number variants and chromatin modifications) could further ameliorate these issues and also provide new mechanistic insights. In this context, systems biology tools that learn the relationships between different layers of biology are needed. Along this line, the release of CCL screens with readouts other than growth inhibition or proliferation rate [90, 91] will help unveil the connections between the genetic background of the cells and the phenotypic outcome of drug treatment.

All in all, transcriptomics is likely to remain the dominant genome-wide data type for drug discovery, as recent technical and statistical developments have drastically reduced its cost [92]. The L1000 Next-generation Connectivity Map, for instance, contains about one million post-treatment gene expression signatures for 20,000 molecules [90]. These signatures await to be interpreted and annotated, and more importantly, they have to be associated with pre-treatment signatures in order to identify therapeutic opportunities. We believe that network biology strategies like the one presented here will enable this connection, encircling relevant "regions" of the signatures and measuring the distances between them.

## Additional files

**Additional file 1:** Contains supplementary figures 1–16. (PDF 2107 kb)

**Additional file 2:** Collection of drug modules in GMT format. The first column indicates the name of the drug while the second column indicates whether the module is a secondary module ("second_module") or not ("na"). From the third column onwards, there are the genes composing the module (gene names). (XLSX 219 kb)

**Additional file 3:** Drug module-gene correlations across tissues. (XLSX 2823 kb)

**Additional file 4:** Pair-wise distances between drug modules. Network distances ($d_{net}$) are normalized (z-scores): negative values denote proximity. Secondary modules receive with the suffix "_md2". See the "Methods" section for a detailed explanation of the network distance measurement. (XLSX 1742 kb)

**Additional file 5:** Enrichment scores and *p* values between drug modules (rows) and Hallmark gene sets (columns). For simplicity, secondary modules were merged with the main ones. (XLSX 453 kb)

**Additional file 6:** Enriched (*p* value < 0.001) drug module count across 31 TCGA cancer types, i.e., number of patients where each module is "expressed". (XLSX 79 kb)

**Additional file 7:** Cancer driver and drug module associations (OR > 2, *p* value < 0.001), based on patients "expressing/not-expressing" a module and "having/not-having" a driver mutation in the TCGA cohort. (XLSX 56 kb)

**Additional file 8:** We have chosen 21 features from network-based measures and other functional data: (i) General features (*columns 2–9*). They illustrate basic and general features derived from the omics panel. We provide, for instance, the number of genes in each module, the average correlation among module genes and a measure of how "unique" are those genes with respect to other modules. Besides, we annotate drug classes and the AUROC predictions in both the GDSC and CTRP panels. (ii) Network features (*columns 10–12*). These include distances between module genes and drug targets, "connectivity" within module genes (i.e., distance between them), and proximity to genes from other modules. (iii) Biological features (*columns 12–21*). A series of biological features related to drug biology. We give most of them as simple proportions of genes/proteins. Among others, we provide the cellular compartmentalization of the genes, transcription factor specificity and the proportion of disease-related and "druggable" genes inside the module. Annotated drug targets are listed as well. (XLSX 156 kb)

**Additional file 9:** CLEAN cluster results using drug module genes and Hallmark gene sets. We provide an additional table with the significant associations between gene clusters and hallmark gene sets. Compressed folder (ZIP). The file can be found at https://figshare.com/s/932dd94520d4a60f076d (ZIP 3220 kb)

**Additional file 10:** Peer Review file. (PDF 1933 kb)

## Acknowledgements
The authors would like to thank Prof. Ben Raphael and Dr. Matthew Reyna (Princeton University) for guidance in the use of HotNet2.

## Funding
A.F-T. is a recipient of an FPI fellowship. P.A. acknowledges the support of the Spanish Ministerio de Economía y Competitividad (BIO2016-77038-R) and the European Research Council (SysPharmAD: 614944).

## Availability of data and materials
Gene expression, drug response and drug target datasets are available in the cancerxxgene resource (https://www.cancerrxgene.org/gdsc1000/GDSC1000_WebResources/Home.html). External validation data for drug response predictions are available in the ctd2 repository (https://ocg.cancer.gov/programs/ctd2-data-portal). Hallmark gene sets were obtained from the Molecular Signature Database (MSigDB) (http://software.broadinstitute.org/gsea/index.jsp). TCGA gene expression data were downloaded from cbioportal (http://www.cbioportal.org; "PanCancer Atlas" flag). TCGA cancer drivers are available in the OncoGenomic Landscapes resource (https://oglandscapes.irbbarcelona.org/). All data generated or analyzed during this study are included in this published article and its additional files. Code for the module detection pipeline is available at https://github.com/sbnb-lab-irb-barcelona/GDSC-drug-modules.

## Authors' contributions
AF-T, MD-F, and PA conceived the study. AF-T and MD-F performed the analyses. AF-T, MD-F, and PA interpreted the data and wrote the manuscript. All authors read and approved the final manuscript.

## Ethics approval and consent to participate
Not applicable.

## Consent for publication
Not applicable.

## Competing interests
The authors declare that they have no competing interests.

## 

## Author details
[1]Joint IRB-BSC-CRG Program in Computational Biology, Institute for Research in Biomedicine (IRB Barcelona), The Barcelona Institute of Science and Technology, Barcelona, Catalonia, Spain. [2]Institució Catalana de Recerca i Estudis Avançats (ICREA), Barcelona, Catalonia, Spain.

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
