## [Peer Review file. (PDF 1933 kb) · Genome Medicine]

Additional file 10

Encircling the regions of the pharmacogenomic landscape that determine drug response

Adrià Fernández-Torras, Miquel Duran-Frigola and Patrick Aloy

Point-by-point responses to Reviewers' comments

First of all, we would like to thank the Editor and the Reviewers for their critical reading of our manuscript and for the positive feedback. We have thoroughly addressed all of the Reviewers' concerns and, we feel, we have now achieved a much better manuscript.

In summary:

- We have performed comprehensive robustness analyses of every step of our pipeline. This has led to the choice of finer thresholds/cutoffs that were previously deemed as arbitrary by one of the Reviewers.
- We have benchmarked our modules in an external validation setting (i.e. using an independent cancer cell line panel).
- We have shown that our module selection procedure has potential for increasing performance of machine-learning algorithms in the task of predicting drug sensitivity.
- Following Reviewers' insights, we have significantly deepened into illustrative examples and added further proof of their validity.
- We have prepared new Supplementary Data to enrich our module characterization, following a suggestion by one of the Reviewers to use a state-of-the-art cluster enrichment analysis method.
- We have quantified the tissue-specificity of our modules.
- We have significantly complemented our exploratory analysis of TCGA samples, including correlations between drug modules and driver genes and tumor types.
- We make our code available as a GitHub repository: <https://github.com/sbnb-lab-irb-barcelona/GDSC-drug-modules>
- We have thoughtfully revised the text, updated figures and added supplementary materials as necessary.

Below, you can find the point-by-point response to Reviewers' comments.

Reviewer #1

The manuscript by Fernandez-Torras et al. presents an approach to derive drug modules from gene expression data, filtering and a network-based method to identify the module itself. There is a nice presentation of feature analysis that the authors leverage in their work, such as looking which genes (including a specific analysis on drug targets) are correlated to drug response and how this relates to drug class and to tissue type. While this part is not novel, it does contribute to the overall story presented. The method itself starts by filtering out commonly correlated genes, then uses GSEA to select genes that participate in annotated reactome pathways. The selected genes are mapped onto STRING and HotNet2 is run to identify network modules and then DIAMOnD was run to expand on each of the modules. From these results the authors present an analysis of the network modules and how they compare to gene expression profiles. They show the modules are predictive and related to the MOA of the drug. Overall, the work is interesting and clearly written. There are few minor questions/concerns that I would like to raise, but I do support the publication of this work.

We thank the Reviewer for his/her positive comments. Below we address each of them point-by-point.

The method uses DIAMOnD to expand modules and gain genes that were not in reactome pathways or are highly correlated. I am curious why this step is needed and what it adds to the process. Are the HotNet2 derived modules sufficient to capture MOA and be predictive of drug response? If not, then why do the added genes provide more information to the predictions? I would like to see a bit more on this aspect of the method.

We added the DIAMOnD step at the end of the pipeline in order to reconsider genes that were significantly correlated but discarded in the Reactome-based filtering step, mainly due to the incomplete coverage of the Reactome database. We would like to emphasize that the DIAMOnD step has anecdotal impact on the modules' composition (only about 5% of the genes come from the DIAMOnD "expansion"). Please see Figure R1 below, and further exploration of the robustness of the DIAMOnD step in response to a related comment by Reviewer #3.

We now more clearly explain the motivation behind the use of DIAMOnD in the main text, and clarify that this step of the algorithm has minimal impact on the composition of the modules.

Figure R1. Number of genes added by the DIAMOND step in relation to genes added by HotNet.

The weakest part of the paper is when the authors identify modules in the TCGA data. They simply state that the modules are there and end the analysis. I would like to see a bit more in terms of looking into the biology of the disease and how it relates to the modules. How do these patterns change across cancer types. Do the authors find what would be expected, such as MAPK signaling drug modules being identified in melanomas where BRAF and NRAS are dominant? How do the modules relate to driver genes? Even though the drug treatment data in TCGA are not great, there is sufficient information to allow the authors to stratify patients on the modules and then look at survival. For instance, stratify all patients treated with drug X on the presence of the associated module. Do the patients treated with drug X and have the drug X module show differential survival?

We agree with the Reviewer that the TCGA study was weak. We are particularly thankful to the Reviewer for his/her intuition about MAPK signaling drug modules, which nicely stands out in our module-based analysis. In Figure R2A, we see that positively correlated modules (PCMs) for drugs affecting MAPK signaling (e.g. Dabrafenib) have a tendency to be “found” (i.e. expressed) in skin cutaneous melanoma (SKCM) patients. Moreover, when we compare MEK and BRAF inhibitors (both are MAPK-related mechanisms), we observe that, in the specific case of melanomas (SKCM), BRAF inhibitors are more detected among patients than MEK inhibitors (Figure R2B). Of note, one PPAR inhibitor (FH535) was found enriched in a high number of SKCM patients, in

good agreement with findings by others suggesting the use of PPAR inhibitors to treat skin cancers [1,2]. We have included this illustrative example in the main text.

Figure R2. (A) Number of modules identified in SKCM patients. Each dot corresponds to a positively-correlated module of a drug, and the y-axis measures the proportion of SKCM patients where the corresponding module is “found”, as defined by a GSEA enrichment analysis ($p \leq 0.001$). BRAF inhibitors are highlighted in red; MEK inhibitors in blue. (B) Similarly, proportion of patients where MEK and BRAF inhibitors are found, this time across tumor types beyond SKCM.

Other illustrative examples, analyzed in a similar way, include (Figure R3):

- Pancreatic cancers (PAAD) and EGFR inhibitors: It is well known that EGFR overexpression is needed for pancreatic tumorigenesis and is common among PAAD tumors [3,4]. Accordingly, the modules of EGFR inhibitors (such as Afatinib, EKB-569, Gefitinib, Cetuximab, etc.) are abundantly found in PAAD patients (Figure R3A).
- Glioblastomas (GBMs) and WNT signaling inhibitors: The most abundant drug modules found in GBM patients correspond to two GSK3 inhibitors (CHIR-99021 and SB216763) and one TNKS inhibitor (XAV939) (Figure R3B). These three drugs are targeting the WNT signaling, in good agreement with recent studies showing promise for WNT-related therapies against GBMs [5,6]. Next to these three drugs, we found Gefitinib (EGFR inhibitor), which entered clinical trials to treat GBM based on the high expression of EGFR in this tumor type [7] (conventional EGFR inhibition has pharmacokinetic issues [8], though). Finally, we found a PLK3 inhibitor (NPK76-II-72-1) that was, in turn, enriched in MYC-related processes (see comments to Reviewer #2). In fact, inhibition of PLKs (and also GSK3 inhibition [9]) has been recently shown to be effective against MYC-activated malignant glioma cells [10].

- Mechanistic heterogeneity in breast cancers (BRCA): The most abundant module in BRCA patients is the one of KIN001-102, an inhibitor of AKT, a kinase that has an important role in BRCA throughout PI3K signaling [11] (Figure R3C). Other relevant drug modules include those of WZ3105 (IRAK1 inhibitor) and NPK76-II-72-1 (the PLK3 inhibitor above). Both IRAK1 and PLK3 have been proposed as breast cancer targets as well [12–14]. These examples thus illustrate the heterogeneity of BRCA tumors [15].

Figure R3. Modules found in (A) pancreatic cancers (PAAD), (B) glioblastomas (GBM) and (C) breast carcinomas (BRCA). Similar to Figure R2A, each dot is a drug module and we measure the proportion of patients where the module can be found.

We have incorporated some of these observations into the main text and updated Figure 4. In addition, we provide a Supplementary Data S6 characterizing the tumor-type specificity of our drug modules in light of the TCGA cohort. Examples like the ones explained above can be mined therein.

Following the Reviewer’s suggestion, we then analyzed the connection between cancer drivers and the drug modules identified in TCGA patients. For this, we used the Catalog of Driver Mutations as detailed in a recent publication by our group [16]. Gene expression data were available for 9,780 TCGA patients; 113 well-known driver mutations could be found in this cohort. We then performed a pairwise driver-vs-module enrichment analysis in order to identify drug modules that were significantly enriched in patients harboring specific driver mutations. We simply measured this enrichment with a Fisher’s test on a contingency table classifying patients as “expressing module of drug X” (p -value < 0.001) and “harboring a mutation in driver gene Y”.

The results of this analysis are now given as a Supplementary Data S7. We identified several compelling module-driver associations ($OR > 2$, p -value < 0.001). For example:

- ERK/MAPK signaling inhibitors: The modules of drugs targeting ERK/MAPK signaling pathway were strongly related to patients having mutations in BRAF (8/11 drugs). Of note, 4 of the 5 MEK inhibitors (AZD6244, CI-1040, PD-0325901 and Trametinib) were characteristically enriched in patients having mutations in HRAS, in line with recent publications showing higher sensitivity of AZD6244 and PD-0325901 when HRAS is mutated [17]. Both BRAF and HRAS are known to have critical roles in the MAPK signaling cascade [18,19].
- EGFR signaling inhibitors: The driver genes most frequently associated to EGFR inhibitors were KRAS (found in 3 of the 5 EGFR inhibitors, namely Afatinib, CP724714 and EKB-569) and BRAF (found in 2 of the 5 EGFR inhibitors, namely Cetuximab and EKB-569). Indeed, crosstalk has been reported between these two driver genes in pancreatic cancers (PAAD) [20]. In turn, the modules of EGFR inhibitors are abundant in PAAD patients (Figure R3A).
- Mitosis inhibitors: 3 of the 6 mitosis inhibitors (GSK1070916, NPK76-II-72-1 and SB-715992) have significant associations with cancer drivers. The three of them are related to mutations in Cyclin D2 and RB1, two major players in cell cycle [21]. Remarkably, the PLK3 inhibitor NPK76-II-72-1 was significantly associated to mutations in MYC, adding evidence to the association that we found between PLK3 inhibition and MYC-related processes (please see our response to Reviewer #2).

Finally, as recognized by the Reviewer, and by others [22], treatment data in the TCGA are of poor quality. Treatment outcomes are not always provided and, importantly, a substantial proportion of patients were treated with more than one drug. Moreover, in many cases, it was not clear whether gene expression data for the tumors was measured at a tumor stage corresponding to the time of treatment. Given the severe limitations of the data, we prefer to avoid claims as to the predictive potential of our modules in the clinics (see response to Reviewer #2 for a more realistic external validation related to an independent cancer cell line panel). Rather, we prefer to use the TCGA as a reference cohort to merely evaluate the “presence” of our modules in a clinical setting. We hope that this choice is justified by the extensive analysis of drivers and tissue associations explained above.

Nonetheless, we were curious to see whether there was an association between TCGA treatment data and drug modules. TCGA treatment data was kindly provided to us by Prof. Eytan Ruppin (in relation to synthetic lethality/rescue studies by his group e.g. [23]). Only 20 of our drugs had treatment data in TCGA (3 of them were given only in combination), corresponding to 916 patients. We tested (i) if modules were identified in patients prescribed with a certain drug, and (ii) if, given a drug treatment, patients “expressing” the drug module were indeed more responsive. We found significant associations (OR > 2, p-value < 0.05) in test “i” for three of the drugs (Bleomycin,

Etoposide and 5-Fluorouracil). As for test “ii”, we unfortunately lacked statistical power to assign significance to our results. Trends were promising (OR > 2), though, for 12 of the drugs. For example, we had one responder and one non-responder to Trametinib; the responder had the Trametinib module “expressed” and the non-responder didn’t. In Table R1 we show how our results are overall significant, suggesting that drug modules could indeed have clinical relevance. Nonetheless, given the low counts (low statistical power), we prefer to argue that these observations are not conclusive, and again, we prefer to avoid claims as to the predictive power of the modules in a clinical setting.

	Significant (OR > 2, P < 0.05)			Promising (OR > 2)		
	Observed	Expected	P-value	Observed	Expected	P-value
(i) Treated vs non-treated	3	1.29	0.12	8	4.85	0.072
(ii) Responder vs non-responder	0	0.19	1	10	6.18	0.007

Table R1. Number of drugs with “significant” or “promising” results in the exploratory TCGA analysis. Test “i” measures whether patients “expressing” the drug module were indeed prescribed with the drug. Test “ii” checks, among patients prescribed with a certain drug, whether the module is “expressed” in the group of “responders”.

All in all, we have substantially updated the part of the manuscript related to the TCGA analysis, making it more clear that our aim was *not* to predict drug treatment outcomes. Rather, we chose to enhance our analyses with case examples related to TCGA tissues and driver genes, so as to better characterize and contextualize the modules. Put together, we hope that the TCGA part of the paper is now stronger and more supported.

Page 6, line 53, the authors state "testify drugs" which is likely supposed to be "test drugs"

Thank you. The text has been modified.

Page 8, line 18, the authors use "being" when it looks like they mean "making" or something similar.

Thank you. We applied the suggested change.

Reviewer #2

Fernandez-Torras et al. present an approach to identify sets of genes that are correlated with the sensitivity of cancer cell lines to targeted therapies. They reuse the data of the Genomics of Drug Sensitivity in Cancer (GDSC) to develop and test a new

algorithm that filter genes correlated with drug response based on prior information of signaling networks. They conclude that their approach allows for a better interpretation of the signature of correlated genes.

The authors perform substantial comparisons, their work is well described, and I congratulate them on their detailed methods and extensive supplemental material. Unfortunately, I have strong reservations about some of the content of the manuscript. My major concern is that the authors have not properly shown if their approach is predictive, neither if it performs to the same level as other similar approaches (independently of interpretability that may be too subjective to be quantified). I also think that the complexity of the algorithm raises questions about some of the arbitrary choices made by the authors. Finally, I found that the insights provided as examples of the application of the algorithm are vague and would thus benefit from further validation or better illustrations.

We thank the Reviewer for acknowledging the description of our work, as well as the detail in the methods and supplemental material. Below, we respond point-by-point to his/her concerns related to the algorithm, its predictive capability and the examples discussed in the text.

Predictive power and algorithm

Testing for a "given drug whether sensitive cell lines were enriched in the corresponding drug modules" cannot be considered as an assessment of the algorithm predictive power as there is no proper definition of a training set and a test set for cross-validation. It is at best an evaluation of quality of the model, but not of its prediction. Although the authors acknowledge that "what matters in this baseline exercise is that the predictive potential is maintained when using modules instead of signatures", they should still properly assess prediction. True predictive power can only be tested on the CTRP lines that were not part of GDSC (i.e. real independent test set). With this test sets, the authors should then compare to correlative signatures or other relying on prior knowledge such as Ferranti et al., Bioinformatics, Vol 33(22), 2017.

The suggestion by the Reviewer does indeed make a better assessment of the "predictive potential" of our modules. While in the previous version of the manuscript we did evaluate the modules in the external CTRP dataset, we did not separately measure performance in the subset of cell lines that were unique to the CTRP. In Figure R4, we now distinguish module performances between cell lines "common" to GDSC and CTRP (gray) and cell lines "unique" to CTRP (blue). As expected, "common" cell lines are in general easier to predict (higher AUROC). Nevertheless, we still achieve reasonable

performances (AUROC > 0.6) with the “unique” cell lines subset for a majority of the drugs. This speaks in favor of the generalization capability of our drug modules. Please note that the CTRP validation has the additional challenge posed by discrepancies between sensitivity measures in different cell line panels (this is an issue that we explored in the context of machine learning in a related publication by our group [24]). Figure 3 in the main text has been updated to include the CTRP external validation proposed by the Reviewer.

Figure R4. External validation of our drug modules. We measure their ability to identify sensitive cell lines (top 25, 50 and 100) in the CTRP cancer cell line panel. Performance is evaluated using the area under the ROC curve (AUROC). Gray boxes correspond to cell lines that are shared between GDSC and CTRP, whereas blue boxes correspond to unique CTRP cell lines.

Then, following Reviewer’s suggestion, we carried out an analysis similar to the one presented in the Ferranti et al. paper. In sum, this paper demonstrates that the use of prior knowledge increases the predictive capability of standard/robust machine learning algorithms such as random forests. To check whether our modules could be used as a pre-filtered (i.e. knowledge-enriched) list of features in such algorithms, we trained random forests for each drug (using parameters as specified in the Ferranti et al. publication), and checked in the “unique” external CTRP dataset whether they were able to uprank the top 25, 50 and 100 most sensitive cell lines. In Figure R5 we show the results (AUROC) of using full gene expression profiles (blue) as features, as well as the results of using only module genes (red) (corresponding to less than 1% of the full profiles). The use of drug modules yielded, in general, better performances.

Taken together, these results show potential for the use of our modules as prior knowledge in more elaborate predictors of drug sensitivity.

Figure R5. Performance of random forest predictors of drug sensitivity (a predictor was built for each drug; predictions for the top 25, 50 and 100 most sensitive cell lines are shown). (Top) Distribution of AUROC for the predictors using full gene expression profiles (blue), and module-specific profiles (red). (Bottom) A paired view of the AUROC values.

Given that major concern, I found that applying this approach on TCGA samples is futile, especially that no comparison is made to other published approaches using TCGA data.

We received major comments about the TCGA analysis by other Reviewers too, and we have now re-structured this part of the manuscript in order to address all of them. In sum, we prefer to constrain our claims about predictive power to the realm of cancer cell lines, and we recognize that good predictions of drug response in a clinical setting pose additional challenges that are out of the scope of this manuscript. Treatment response data in the TCGA are of poor quality [22] and, for the most part, correspond to therapies that involve more than one drug and/or do not necessarily match the time-point of collection of gene expression data. Instead, we looked for the “presence” of our modules in TCGA tumor samples, so as to ensure that modules could be identified in patient data and were not unique to cancer cell lines. Then, we checked whether drug modules could be associated with certain tumor types and driver genes in the TCGA clinical cohort. Results of this analysis are now enclosed as Supplementary Data S6 and S7. We have written an extensive explanation about TCGA issues in response to

Reviewer #1 (including Figures R2 and R3). We kindly refer the Reviewer to those answers for further details.

I also think that the authors should justify why using Reactome to filter genes improves the outcome of the HotNet step (which is also meant to filter genes based on prior knowledge); and why the modules identified by DIAMOND are more informative than the Reactome enriched sets themselves.

The HotNet algorithm was originally developed to deal with mutation data [25], which usually involves a smaller number of genes compared to transcriptomics data. We found that running HotNet *directly* after the gene-drug correlation analysis yielded worse signal-to-noise ratios than those explored by the HotNet authors. For this reason, we chose to pre-filter the HotNet input using the Reactome knowledgebase with the hope that, upon filtering, the input of HotNet would be more circumscribed within certain regions of the interactome. In Figure R6A, we show two exemplary HotNet curves corresponding to the Afatinib drug module detection. The steeper the curve, the “earlier” the module is detected as we “grow” it. In the left panel, we show that, *without* Reactome-based filtering, the module identification for the observed (red) data does not markedly differ from randomized runs (blue). In contrast, if Reactome is used to filter the HotNet input (right panel), the algorithm more clearly distinguishes between observed and random inputs. The difference between the observed and the random HotNet runs can be quantified by subtracting the respective areas under the curves (diffAUC). In Figure R6B we explore this systematically across drugs. It can be seen that using Reactome filtering increases the diffAUC approximately two-fold with respect to the “vanilla” (non-prefiltered) HotNet runs.

Figure R6. (A) HotNet statistic for the positively-correlated module (PCM) detection in Afatinib. Red lines correspond to observed data and blue lines to random runs. In the left panel, HotNet was run *without* pre-filtering. In the right panel, a Reactome-based pre-filtering was applied. (B) Difference between observed and random HotNet curves

(quantified as the subtraction of areas) across all drugs, with and without the Reactome filtering.

As for the DIAMOnD step, we decided to add it at the end of our pipeline in order to reconsider genes that were significantly correlated to the drug *but* excluded by the Reactome-based filtering, mainly due to the incomplete coverage of the Reactome database. In Figure R1 (see answer to Reviewer #1) we demonstrate that this step has a minimal impact on the module composition, as only 2 to 7 genes (5%) in the modules are included upon DIAMOnD module “expansion”. In the new version of the manuscript, we make it more clear that DIAMOnD module expansion does not have a critical importance and, additionally, we provide further robustness analyses of the DIAMOnD step (see Figures R21 and R22 in response to Reviewer #3).

Insights into drug mechanism of action

The authors claim to “yield a novel, rich portrait of drug activity” and insist that their results “annotated [drugs] well beyond the reductionist (target-centered) view.” First, I think that the authors should acknowledge that it is well accepted that determinants of drug response go beyond the targets of the drug. Although it is clear that target-centered therapies are dominant in the clinic, basic researchers are most aware of the discrepancies between drug target(s) and sensitivity factors.

We completely agree with the Reviewer and we have modified the abstract accordingly. It now reads: “[our analysis] contributes to a better understanding of the molecular determinants of drug activity.”

Regarding the insights provided by the enrichment of Hallmark gene sets, the overlap of genes between Reactome gene sets and Hallmark gene sets is likely to help the results obtained in fig. 4B and 4C. I think that a fair comparison would be to apply GSEA on Hallmark gene sets with inputs being either the modules (as now), or correlated genes *filtered* by genes present in Reactome (i.e. not performing the filter by GSEA, but removing genes with unknown biology as defined by presence in Reactome). Optionally, the authors could also compare the results (Hallmark gene set enrichment) after GSEA filtering and after HotNet expansion in order to assess the impact of each step of the algorithm on interpretability.

It is true that the gene universe of Reactome may help our Hallmarks enrichment scores if compared to a more “uncharacterized” background. To ensure that the trends observed still hold when we take this fact into consideration, we repeated the Hallmarks enrichment calculations taking the Reactome universe as a reference. Figure R7 demonstrates that, while Reactome-filtered correlations do indeed have a (slightly) higher tendency to yield enrichment signals, the signal obtained with the module-based analysis is still substantially stronger. For simplicity, in the new versions of Figures 4B

and 4C (main text) we now plot the results of this analysis proposed by the Reviewer, instead of the less stringent one previously used.

Figure R7. See Figure 4 legend in the main text. “Correlations (with Reactome)” refers to the reference proposed by the Reviewer.

Finally, it feels like the examples provided are being 'cherry-picked' (e.g. why only some of the RTK drugs are plotted in Figure 4?) and I would expect some of the claims to be validated, or at least better illustrated such that the reader can feel more confident about the observations that "recapitulate non-trivial links between drug classes". For example:

We would like to thank this Reviewer for the feedback about the examples provided. We have addressed the criticisms and we hope that he/she will now be more convinced by our observations. The new version of the manuscript contains updated/augmented figures and text following Reviewer’s suggestions.

As for the RTK drugs shown in Figure 4, we only include the ones for which we could identify modules (see Supplementary Data S1) (no modules could be found for CH5424802, PD-173074, XMD14-99 and OSI-930). The issue of drug coverage is now more explicitly commented in the text.

Please note here that, following comments by another Reviewer (#3), some cutoffs along the pipeline have been modified and the manuscript figures correspondingly updated, including the ones discussed below. We kindly ask the Reviewer to refer to the updated figures when considering our responses to his/her comments.

Moreover, we have added the significance values (hypergeometric test) in the new version of Figure 4A, on top of the odds ratio, in order to be more rigorous with our claims, especially when counts were low.

"processes associated to the Myc transcription factor were enriched in all of the drugs, suggesting that the efficiency of mitosis targeting depends on the status of this proto-

oncogene." This is somewhat expected, but an illustration of the main genes driving this enrichment would be valuable. In addition, the author should test is most of this signal is driven by Myc amplified/overexpressing cell lines.

In Figure R8 we show three drugs whose modules were significantly enriched (p-value < 0.01) in MYC signaling, namely NPK76-II-72-1 (MYC1&2), GSK1070916 (MYC1) and MPS-1-IN-1 (MYC1&2). The three drugs had a rather distinct module composition in terms of MYC-related genes. NPK76-II-72-1 (the drug with the highest enrichment of the MYC Hallmark gene sets) contained most of the MYC-related genes. The MYC gene itself was unique to the MPS-1-IN-1 module, together with the CLNS1A and RPS6 genes. Finally, CUL1 was the only MYC-related gene unique to the GSK1070916 module; the rest were shared with the NPK76-II-72-1 module (e.g. USP1, FBL, TRA2B, etc.)

Figure R8. Detailed view of the MYC-related genes contributing to the enrichment signal reported in Figure 4A (main text).

Following Reviewer's comment, we then checked whether MYC expression levels correlate with sensitivity to these three drugs. In Figure R9 we show the results of a GSEA-like enrichment analysis where cell lines are sorted by their level of MYC expression, and the top 25, 50 and 100 most sensitive cell lines for each drug are evaluated for their tendency to having high MYC expression levels. We observed a significant association between sensitivity (top 50 and top 100) and MYC expression levels for the MPS-1-IN-1 drug (the drug that, in fact, had MYC as a gene in the module). For GSK1070916 (the drug that had the MYC1, *but not* the MYC2 gene-set significantly enriched in the module), we observed no correlation between MYC expression levels and sensitivity. Finally, for NPK76-II-72-1 we observed a significant enrichment when considering only the top 25 sensitive cell lines, but the signal vanished for the top 50 and top 100 cell lines.

We present these results as a Supplementary Figure S15 and mention the findings in the text.

Figure R9. GSEA-like enrichment analysis of sensitive cell lines (top 25, 50, 100), ranked by their level of MYC expression. Three drugs having modules enriched with MYC-related genes are analyzed.

- "drugs targeting [the IGFR] pathway were proximal to cell replication processes such as mitosis, cell cycle, DNA replication and other nucleus-related events." How IGFR inhibitor differ from other RTKs, is this relation driven by specific genes or specific cell lines?

To address this question by the Reviewer, we defined a "pharmacogenomic" metric that measures the similarity of cell sensitivity profiles between two given drugs. This was simply done by calculating the Pearson's correlation between the sensitivity (1-AUC) profiles of the two drugs. In Figure R10A we plot the pharmacogenomic distance *within* and *between* RTK and IGFR inhibitors. It is apparent from this plot that IGFR inhibitors cluster together and differ from RTK signaling inhibitors. RTK inhibitors, in turn, show 2-3 clusters. When zooming in on these results, we found that RTK inhibitors characteristically affect cell lines belonging to a varied array of tissues (large intestine, aerodigestive tract, skin, etc., e.g. COLO-205, TE-11, NCI-H2122, VCaP, KS-1, HMV-II and HT-29), whereas A204, G-402 (soft tissue) and CGTH-W-1 (thyroid) cells lines are affected by IGFR inhibitors.

Since our module-detection pipeline capitalizes primarily on sensitivity profiles, we must attribute the difference between IGFR inhibitors and RTK inhibitors to the differential sensitivity across cell lines. A similar pattern is reflected at the gene/network level (Figure R10B), which makes it difficult to discern between "cell line" and "gene" contributions to the difference between the two drug families.

Figure R10. (A) Pairwise comparison of RTK (purple) and IGFR inhibitors (red) based on the pharmacogenomic distance. Drugs are hierarchically clustered. (B) Pairwise comparison between the same drugs, this time using the network-based distance of the modules (see Methods in the main text).

- "a subgroup of RTK inhibitors (namely ACC220, CEP-701, NVP-BHG712 and MP470) were characteristically associated to the PI3K-AKT-mTOR signaling cascade." It may be true that FLT3 - the shared target - is connected to the PI3K/mTOR pathway, but the authors should discuss why drugs targeting ErbB2 or PI3K directly are unexpectedly less connected among themselves than with FLT3 inhibitors.

We thank the Reviewer for this observation, which we believe comes from an inspection of Figures 3C and 4A. In our dataset, there are 18 drugs targeting the PI3K/mTOR signaling pathway. As correctly pointed out by the Reviewer, the average "similarity" (as measured by network proximity) of the drug modules in this category is relatively low (Figure 3C). Closer inspection of this statistic reveals that, in fact, PI3K/mTOR signaling inhibitors comprise different mechanisms of action (PI3K, mTOR, AKT and PDK1 inhibitors). When we analyze the distance between the modules of these drugs, several clusters are revealed (Figure R11A), illustrating the heterogeneity of this class of compounds. Interestingly, ERBB2 inhibitors cluster together and separate from PI3K/mTOR signaling inhibitors. On the contrary, and in line with the Reviewer's observation, FLT3 inhibitors are rather widespread and have some notable "proximities" to PI3K/mTOR signaling inhibitors such as GDC0941 (PI3K inhibitor). For completeness, we also provide the pharmacogenomic distance between these drugs (Figure R11B), showing similar trends.

Moreover, when we looked for enrichment of the PI3K/AKT/mTOR pathway in modules corresponding to ERBB2 inhibitors and FLT3 inhibitors, we reassuringly found modules of ERBB2 inhibitors (Afatinib, CP724714 and CUDC-101) to be enriched in this pathway, as well as the modules of 3 of the 6 FLT3 inhibitors (AC220, CEP-701 and MP470) (Figure R11B). Interestingly, we did *not* find this enrichment for many of the drugs targeting the PI3K signaling pathway directly.

Figure R11. Similar to Figure R10, (A) network-based distance between modules of PI3K/mTOR signaling inhibitors, ERBB2 inhibitors and FLT3 inhibitors. Pharmacogenomic distances between drugs are shown in (B). (C) Enrichment analysis of the drug modules of these inhibitors performed against the “PI3K/AKT/mTOR signaling pathway” geneset.

The paragraph "Compared to the two previous drug classes, the modules of ERK-MAPK pathway inhibitors are enriched in a wide range of biological processes (Figure 4A, bottom)." is vague and not conclusive. One important aspect to address is how the cell line AUCs are correlated between drugs with the same nominal target. If this is the case, why are the enrichment modules different? If this is not the case, the author could identify which cell lines differ and illustrate how that the disparities in enriched Hallmark gene sets explain the dissimilarities in sensitivity.

It is true that this sentence was vague and not conclusive. It now reads: “As for ERK-MAPK pathway inhibitors, we could observe a total of 17 enriched Hallmarks, making this class of drugs the one with more variability in terms of enrichment signal of the modules”. This statement is supported by Figure R12A where drugs targeting ERK-MAPK signaling stand out (together with drugs targeting WNT signaling) as having a large number of enriched Hallmarks (17 on average) compared to e.g. IGFR and PI3K/mTOR drugs (5 and 7 on average, respectively).

Following Reviewer’s suggestion, we have calculated the pharmacogenomic distance between the drugs contained in the ERK-MAPK signaling MoA category. The pairwise analysis of sensitivity profiles reveals two large clusters, corresponding to MEK inhibitors (large cluster) and BRAF inhibitors. Three of the drugs (FMK, VX-11e and HG-6-64-1) were rather distinct both in terms of nominal target and/or sensitivity profiles. Among the cell lines characteristically sensitive to MEK inhibitors we found BPH-1, LB2518-MEL, OCUM-1, SK-CO-1 and CL-11 (urogenital system, skin, digestive system and large intestine). On the contrary, BRAF inhibitors were potent against cell lines such as Hs940-T, 8-MG-BA, SNG-M, NMC-G1 and T-24 (skin, nervous system and urogenital system).

Figure R12. (A) Number of enriched Hallmarks found in drug modules, organized by mechanism of action. (B) Pharmacogenomic distance of RSK, ERK, MEK and BRAF inhibitors.

Concerning the MPS1 inhibitor: are the most sensitive lines of medulloblastoma origin? Is the enrichment driven by such lines, or does the finding arise from multiple lines and can then be related to medulloblastoma (which would be a good illustration of the algorithm capability)?

We had 4 medulloblastoma cell lines in our dataset (please see filtering procedures in the main text, Methods). In Figure R13 we show the potency of the MPS1 inhibitor (MPS-1-IN-1) across the cell line panel. Clearly, medulloblastoma cell lines (in red) are not among the most sensitive ones.

We decided to remove this example from the manuscript because, after re-running our analysis with small modifications suggested by other Reviewers (mainly Reviewer #3), we lost the enrichment signal of Hedgehog signaling, suggesting that this example was not robust and strong enough. The rest of the examples explained above were robust to the fine-tuning of the pipeline.

Figure R13. Cell line sensitivity to the MPS-1-IN-1 drug. Medulloblastoma cells are highlighted in red.

Other concerns:

"we first looked for artifactual associations between drug responses and CCLs, depending on their tissue of origin." The authors should address the artefact related to nominal cell division rate as illustrated in Hafner et al., Nat Biotechnol. Vol 35(6), 2017 and account for it as it has been done in Haverty et al., Nature, Vol 533(7603), 2016.

We recognize that correcting for nominal cell division rate is a recommendable procedure, especially when different cancer cell line panels are to be compared. We did not initially do so because our aim was to build upon a seminal drug-gene correlation

paper [26] that dealt directly with sensitivity curves, as well as the original GDSC publication [27] that already implemented a robust metric (the AUC) having desirable properties such as correlation with mechanism of action and ability to be fitted by relatively simple models.

Nonetheless, we tried to address this comment by the Reviewer in order to ensure that our results were robust to the metric used as a drug response measure. In particular, we tried to convert 1-AUC (area over the curve, AOC) measures available from GDSC to growth-rate corrected AOCs (GRAOC) following the methodologies suggested by the Reviewer. Others have tried to achieve so recently by inferring division times with gemcitabine:

<https://github.com/bhklab/ConsensusPGxPaper/tree/master/AlternativeDrugResponseMetrics>.

Unfortunately, the procedure could only be applied to 126 cell lines, which is a substantially smaller number compared to the original panel (of note, [26] recommends the use of at least 400 cell lines for drug-gene correlation computation).

Despite the low coverage in terms of cell lines, we re-ran our full pipeline considering the AOC (original metric) and GRAOC sensitivity measures across the 126 available cell lines. In Figure R14, we can observe that approximately 40% of the drug-gene correlations remained when using GRAOC instead of AOC and vice versa (A), and that, after the Reactome-based enrichment, almost 60% of the pathways detected were the same (B). More importantly, when we measured the network-based distance between modules identified with either metrics, we found that they were remarkably proximal in the network (C). This demonstrates that, even if differences exist in terms of gene composition of the modules, modules are still circumscribed within the same “regions” of the interactome when the drug response metric is changed.

Figure R14. Impact of the GRAOC and AOC metrics on the pipeline (positive correlations, red; negative correlations, blue). (A) Proportion of genes found in common in the drug-gene correlation step. (B) Similarly, proportion of enriched Reactome

pathways. (C) Network distance between AOC-GRAOC modules *within* drugs and *between* drugs.

"We noted that the molecular targets for these pathways are usually cell surface receptors, e.g. EGFR, IGFR, ALK, KIT, MET and PDGFRA." This is likely due to the overexpression of the receptor and the author should discuss that.

The Reviewer is perfectly right and we explain this better in the new version of the manuscript: "this trend is mostly driven by the over-expression of the target in the cell surface".

To support this statement, we now provide a Supplementary Figure S5 (Figure R15 below) showing the correlation between target expression and drug response in cases where the drug target is a cell surface receptor (e.g. PDGFRA, ERBB2, EGFR and IGF1R) (Figure R15A). As predicted by the Reviewer, a positive correlation can be observed. We would like to also emphasize that, for the 4 drugs analyzed in Figure R15B, cell lines exist that are very sensitive but do *not* highly express the target. This finding is in good agreement with previous work in our group [28], and similar findings by others (e.g. [26]), reporting that drug target expression is not enough to predict sensitivity. Also, this goes in line with the main rationale of the present work (and many others) as it demonstrates that, even in relatively "simple" cases, the underlying determinants of drug response may be quite complex and impossible to capture with the analysis of individual genes.

Figure R15. (A) Correlation between drugs and cell surface targets. (B) Four exemplary drugs whose nominal target gene expression correlates with cell line sensitivity.

In the section discussing tissue-related effect, in particular when the authors "verified that none of the tissues had a dominant effect on the measures of drug-gene correlations", it unclear if the authors address the case of drugs that are active in only one or few tissues and not in other tissues; this would lead to tissue-specific genes to be the most correlated genes, which means that response signature may not bring information beyond tissue of origin (see comment about MPS1 inhibitor above).

As correctly pointed out by the Reviewer, there are cases where the drug modules are mere surrogates of the tissue of origin. These cases correspond to drugs that are specific to a certain tissue. While these signatures are somewhat less interesting than the signatures that "go beyond" the tissue level, they could still be of interest because of their predictive potential (for example), and this is why we chose to keep and report them. It is true, however, that it would be informative to distinguish between "tissue-specific" and "general" modules. We have added a new Supplementary Data S2 reporting the tissue specificity of each of the modules. This was measured calculating the drug-gene correlations using only cell lines belonging to the tissue in question. Please see our response to Reviewer #3 (Figure R23) for further insights about the tissue specificity of our modules.

The authors should provide a table with the drug targets as used for producing table S4.

We now provide the drug targets in the Supplementary Data S8. This table is accompanied by a caption explaining in detail the drug target annotation procedure.

Drug targets could also be reported in Fig 4a for better understanding of the results.

Thanks for the suggestion. Figure 4A has been modified accordingly.

Reviewer #3

The authors have undertaken a commendable approach to render cell-line sensitivity profiling data more interpretable, which is an issue that has become more clear as more of these datasets emerge. The authors' approach is one of several emerging approaches that aims to leverage prior information to interpret connections between genes that are co-ordinately linked to small-molecule sensitivity.

In general, the manuscript is presented in a logical sequence, and the analyses are well-motivated, in the sense of understanding the problem and presenting a cogent approach to addressing it. Moreover, the authors' scholarship, understanding of the available datasets and the original papers describing them, is quite high.

While we are generally very positive about the approach and the manuscript, we feel that a revision is warranted before publication, including three major areas: (1) controlling and parameterizing computational analyses to address the robustness of results and justify (or eliminate) arbitrary analysis choices; (2) clarifying results and revising assertions about implications that might not be supported by the data; and (3) general revision of narrative and figure legends for clarity, English idiom, and grammar/spelling issues.

We thank the Reviewer for the positive comments about our approach and manuscript. Below, we address all of his/her comments point-by-point.

Detailed comments in each of the three areas follow:

(1)

The choice of keeping a maximum of n correlated genes seems restrictive and antithetical to the idea of modules -- there should be at least n (and probably many more) individually correlated genes, since each mechanism of action could correspond to a whole module. At a minimum, explore what happens to the overall procedure when varying this parameter systematically to be more and less permissive.

We agree with the Reviewer and we apologize for not having explained this part of the algorithm with sufficient clarity. We did not set a fixed cutoff to the maximum number of genes to be allowed in the modules (see, for example, Figure 3A showing that some modules are in fact considerably large). The principal restriction that we applied was a drug-gene correlation score so as to ensure that every gene in the module could be independently “claimed” as being correlated to the drug (as a matter of fact, this score has been fine-tuned following further comments by the Reviewer; see below). It is true, however, that the DIAMOnD-based expansion of the modules *did* have a fixed cutoff. We kindly refer the Reviewer to Figure R1 (in response to Reviewer #1) where we show that the contribution of the DIAMOnD step to the module composition is only marginal.

Throughout our responses below, the Reviewer will find systematic exploration of parameters. We have updated the parameters when applicable, and modified figures and text accordingly.

Similarly, isn't it possible for multiple modules to emerge from the same compound, including modules that have correlations in each direction? How is this explored? How are these connections distinguished during module-finding?

In effect, it is possible that multiple modules emerge from the same compound. To explore this possibility, we ran HotNet iteratively, as HotNet only detects one module per run. In Figure R16A (taking positively- and negatively correlated modules

separately) we can see that, for approximately one third of the drugs, a second module could be found, and that a third or fourth module could be detected only for a small number of drugs. For this reason we considered up to two modules.

As for the “direction” of the modules, we decided to keep them separate to ease interpretability and, more importantly, to make them easier to couple with standard enrichment analysis tools such as GSEA, which are direction-specific in their default modes. We were curious to check, however, if positively- and negatively-correlated modules (PCMs and NCMs) of a given drug were indeed “located” in similar regions of the interactome, suggesting that they belong to a major, composite module. To address this question, we computed network-based distances between PCMs and NCMs *within* and *between* drugs (see [29] and Methods in the manuscript for details about the distance metric). In Figure R16B, it can be seen that PCMs and NCMs do *not* tend to be more proximal within drugs than between drugs (background); they are, if anything, slightly further apart. This observation supports our choice to keep PCMs and NCMs separate.

We have modified the text to better explain this part of the method and we enclose Figure R16 as a Supplementary Figure S10.

Figure R16. (A) Number of modules identified per drug (PCMs in red, NCMs in blue). (B) PCM-vs-NCM distances *within* drugs and *between* drugs.

There are a lot of choices based on $z_c \pm 3$, but it is likely that these distributions are not Gaussian. The authors should address this issue, including use of robust statistics, other distribution models, non-parametric choices, etc.

We agree with the Reviewer that the choice of $z_c \pm 3$ should have been systematically explored, as this parameter may critically influence the composition of the modules. Even if the distribution of Fisher’s z-transformed Pearson’s correlations “tends to normality rapidly as the sample is increased” [30], in practice we did not make this

assumption and considered the empirical distributions instead. In Figure R17A, the Fisher's z-transformed Pearson's correlations (z_c) are shown. We plot the absolute z_c for the left and right tails, corresponding to the observed and shuffled (random) drug profiles. We can see that a z_c cutoff of 3.15-3.25 corresponds to well-accepted/standard "p-values" in one-tailed tests (i.e. $p \sim 0.001$ for the randomized distribution and $p \sim 0.05$ the observed one considering both tails of the distribution). Thus, we chose 3.2 as a justifiable cutoff (see next question by the Reviewer for a robustness analysis of z_c).

One minor point where we did make an assumption about the the distribution of the data corresponds to Figure S3 (Figure R17B in this letter). There, we calculated the mean and standard deviation of the 1-AUC sensitivity measures in order to demonstrate that it is "easier" to detect drug-gene correlations for drugs having a broader dynamic range of sensitivity. We now use robust statistics for this figure, namely the median and the median absolute deviation (MAD).

Figure R17. (A) Absolute z_c for the observed and randomized gene-drug correlations. The chosen cutoff was 3.2 (based on "p-values" of 0.05 and 0.001 observed in the observed and randomized distributions, respectively). (B) Please see Figure S3 legend in the Supplementary Materials. Mean and standard deviation have been substituted by median and MAD.

Finally, please note that p-values derived from randomized/bootstrap analyses (such as the significance of drug targets within modules, Figure 1D) were all empirical and, as such, no assumption was made about the shape of the distribution.

Moreover, the authors should address what happens to their pipeline and its output when less- or more-stringent choices are made for these cutoffs.

Following Reviewer's suggestion, we have performed a systematic assessment of the robustness of our method to the choice of z_c . In brief, we have run our pipeline over a wide range of z_c values [2.5-3.5], and analyzed the composition of the modules

identified. As it can be seen in Figure R18 (top panels), modules become smaller as the *zc* cutoff increases (as expected). Importantly, though, the shrinkage corresponds to a subsetting of the modules, and not to a noisy detection of different gene sets. This can be seen in the bottom panels of Figure R18. For example, over 80% of the genes that were identified with a *zc* cutoff of 3.5 were also identified with a cutoff of 2.5, whereas only about 20% of the genes identified with *zc* = 3.5 were novel if compared to *zc* = 2.5. When focusing on a more relevant region of this screening (e.g. [2.9-3.3]) we can see that ~95% of the genes at *zc* = 3.3 were also present at *zc* = 2.9, whereas only ~5% of the genes were novel).

Figure R18. (Top panels) The black line denotes the number of drug-gene pairs encountered in modules (PCMs and NCMs) as a function of the *zc* score cutoff [range 2.5-3.5]. The continuous colored lines show the number of drug-gene pairs “conserved” in the modules as we move to higher *zcs*, with respect to the cutoff specified in the legend. On the contrary, the dashed lines denote the genes that are added. (Bottom panels) Normalized version of the top panels, taking the total number of drug gene pairs (the black line) as a reference (100%).

The choice of ranking CCLs by sensitivity and then picking top *n* 25, 50, or 100 feels arbitrary, and it's not clear how varying this parameter affects the outcomes. At a minimum, explore what happens by varying this parameter systematically and have some way to judge and report which choices are "good" -- e.g., is there a range of values for which the found modules are stable?

We agree with the Reviewer that, when evaluating the “predictive” potential of our modules, the choice of top 25, 50 and 100 cell lines may feel arbitrary. We chose these cutoffs because, when looking at the number of “sensitive” cell lines assigned by the GDSC authors [27], 25, 50 and 100 approximately corresponded to three critical percentiles of the distribution (Figure R19A). In this same distribution, it can be seen that the number of sensitive cell lines per drug is rather variable. In order to make drugs comparable between them, we preferred to use common thresholds across drugs and see the prediction task as a “ranking” exercise, rather than a classification one. Also, this choice was convenient to perform an external validation with an independent cell line panel (CTRP), where sensitivity labelings were not available. Binarization of sensitivity data is a somewhat complex procedure that is panel-dependent and can be performed in several ways (e.g. [31]), usually considering the IC50 metric instead of the 1-AUC. For these several reasons, we preferred to explore the performance of our rankings considering a more simple scheme, i.e. top 25, 50 and 100 cell lines for each drug.

As suggested by the Reviewer, we explored the performance metric (AUROC) at different cutoffs, ranging from 1 (top cell line) to 150 (top 150 cell lines). In Figure R19B we show the results. As expected, performance decreases as we try to recall more cell lines. There is a sharp decrease in performance from 1 to ~25 cell lines, and then the trend slowly decreases from AUROC ~ 0.8 to AUROC ~ 0.7.

Figure R19. (A) Number of sensitive cell lines per drug according to the GDSC publication. (B) Predictive capability (AUROC) of the top-n sensitive cell lines, n ranging from 1 to 150.

What about when removing genes with low dynamic range of expression first? Only a single way of doing this is proposed. What happens if you change this dynamic-range or expression cutoff systematically. Are the found modules robust to this parameter?

As pointed out by the Reviewer, we only proposed a single way of removing genes with a low dynamic range of expression (low expression values across the cell line panel). We chose an absolute expression value of 5 as a cutoff in the previous version of the manuscript, roughly based on the expression distribution across all genes and cell lines (Figure R20A). We re-inspected this distribution and now found that a cutoff of 4.4 is easier to justify in light of the distribution's shape (4.4 corresponds to the “elbow” separating the population of “unexpressed genes” (left peak) from the “expressed” ones (right distribution)). We confirmed that this choice was robust by measuring the number of drug-gene correlations identified later on. In Figure R20B, it can be seen that, at the chosen correlation cutoff of $z_c = 3.2$ (blue), the number of identified drug-gene correlations is essentially constant around the chosen expression cutoff of 4.4.

Figure R20. (A) Distribution of gene expression values across the GDSC cell line panel. (B) Number of correlations identified after applying different cutoffs for “unexpressed” genes [range 2-10]. The colors of the lines denote the subsequent z_c scores (see Figure R17A). Continuous lines correspond to positive correlations and dashed lines to negative correlations.

There is a fair bit of argument that lineage-specific analyses are not providing additional information, which runs counter to some other studies. What happens if lineage-specific modules are sought? The authors seem not to have explored this because of their findings comparing single-feature correlations.

This is an interesting idea that we only explored superficially in our original analysis, mainly because our goal was to identify “generic” drug modules. We were aware of the fact that some drug modules may be mere surrogates of tissue of origin; this would be the case for drugs that specifically act against a certain tissue. In order to quantify this trend, we generated tissue-specificity plots (new Figure S6). In brief, the aim of these

plots was to check whether drug-gene correlations could be attributed to a certain tissue or, on the contrary, they were “spread” across tissues. We saw examples of both cases. It is true that we put the accent on our ability to identify tissue-independent correlations, which is misleading. We have modified the text so that it is now clear that we capture tissue-specific correlations as well as tissue-agnostic ones.

Additionally, we now include Supplementary Data S2 analyzing the tissue specificity of each drug module (please see our answer to a related comment by Reviewer #2).

Similarly to the above comments, some of the randomization choices during the module-preparation process feel arbitrary and inconclusive. For example, taking 10 random modules of each size, distributing them into a range of intervals, taking the top 200 or 100 genes closest to the module, etc. It looks like the authors were trying to make systematic some of their analyses, but it wasn't clear exactly how they were judging which parameter choices were better, or whether module-formation was stable under different choices. The authors should address these issues explicitly with control experiments.

We were indeed trying to systematize our pipeline, and we recognize that some steps were ill-explained in the previous Methods. We have substantially reworked this section and we hope that our choices are now more clear.

As correctly pointed out by the Reviewer, in the DIAMOnD module “expansion” step. We considered up to 200 genes in the proximity of the module identified by HotNet. 200 genes was proposed by DIAMOnD authors as an optimal cutoff in module expansion based on orthogonal functional analyses [32] (and using the same interactome than ours). We shall recall here that, of these 200 genes, only those that were significantly correlated (first step of the pipeline) were regarded as candidates for incorporation to the final module. In practice, the DIAMOnD step had a minor impact on the module composition (2-7 genes per module, i.e. ~5% of the total module size) (please see Figure R1 in response to Reviewer #1). When we doubled the distance of DIAMOnD exploration to 400 genes, only 2-3 new genes were incorporated per module.

To complement this answer to the Reviewer, we explored the other key parameter of DIAMOnD, namely the alpha coefficient, within a reasonable range [1-10] according to the algorithm publication. As it can be seen in Figure R21, here again the number of genes added to the modules by DIAMOnD does not appear to be affected by the alpha parameter, and we chose to use the default value [$\alpha = 10$].

Figure R21. Number of genes added by DIAMOnD to the modules, using different alpha values (PCMs, left; NCMs, right).

As for the random sampling of modules of given sizes, we added this procedure so as to render the standard network-distance proposed by Menche et al. [29] independent of the size of the modules (i.e. to derive a z-score of the distance). We apologize to this Reviewer for a mistake in our manuscript: we sampled 100 random pairs of modules per pair size, and not 30 as previously written. This is now corrected. Either way, in Figure R22 we demonstrate that the two statistics that are needed for deriving a z-score (i.e. the mean and the standard deviation) are robust to the number of random samples.

Figure R22. Mean and standard deviations of the distance between randomly sampled pairs of modules. Bags of 10-200 pairs of modules were sampled (see Methods for more details about the sampling procedure).

The authors should compare their method to other methods that use prior information to help contextualize drug-response profiling data and other expression signatures, especially ACME analysis from reference 7 and CLEAN analysis: <https://www.ncbi.nlm.nih.gov/pubmed/19640299>.

Thanks for the suggestion. ACME and CLEAN analyses have enriched, reinforced and complemented our results:

ACME

ACME analysis uses cell line features (histological (tissue) and mutation data) as well as drug features (drug targets), together with drug sensitivity data, in order to identify correlations between the features of the cell lines and the features of the drugs. For instance, one can find target-mutation and target-tissue associations. This is indeed a nice complement to our analysis, which is rather focused on finding correlations between drug sensitivity and gene expression levels and does *not* require previous annotations of the drugs and the cell lines.

Interestingly, we could find correspondences between ACME results and ours. For example, we realized, when addressing this comment by the Reviewer, that one of the examples that we highlighted in the main text (i.e. EGFR inhibitors in correlation to EGFR expression; Figure 1) was in fact commented in the ACME paper in the context of the aerodigestive tract (available in our dataset) and squamous cell carcinomas (removed by our filterings). Interestingly, we also captured a related example discussed by the ACME authors regarding ERBB2 inhibitors and ERBB2 overexpression in the breast. We have added these observations as a Supplementary Figure S6C (Figure R23 below), and we now cite the ACME paper in this part of the text.

Figure R23. Please see Figure 1 legend in the manuscript for details. The tissue-specificity of some examples discussed in the ACME paper are highlighted here in light of our results.

CLEAN

CLEAN takes as input a (hierarchically pre-clustered) matrix (e.g. gene expression profiles of cells) and a collection of biological sets (e.g. GO, KEGG). It then dynamically identifies clusters that are highly enriched in any of the genesets. We found that an analysis of this kind could enhance our simpler Hallmark-based enrichment analysis. For this, we downloaded the CLEAN R-package and fed it with our drug modules. The corresponding visualization of the matrix by js.treeview is now provided as Supplementary Data S5 for those readers who are willing to more deeply explore the modules. A simplified version of this table is supplied as Supplementary Data S4, and CLEAN is cited in the text.

Since the idea of a drug-response module is so central to their study, the authors should more carefully distinguish the idea of profiles (all gene expression in a sample), signatures (all differentially expressed genes between samples), and modules (compact signatures filtered and refined with prior information -- in this case the contents of a network biology or pathway corpus).

This is a very good suggestion for nomenclature (profiles/signatures/modules) and we now use it throughout the manuscript.

The authors should clarify the statement "207 drugs with an annotated target" (page 6, line 4): what was the source of target annotations for small molecules?

Following this and other comments by the Reviewers, we have added a Supplementary Data S8 with target annotations for all drugs. In addition, in graphical elements such as Figure 4A we now incorporate the target annotations.

The authors should clarify how they were "Encouraged by the robust predictive power of the drug modules, ..." (page 10): in the previous paragraph, they stated that 1/4 drugs had good predictions, and 1/2 had acceptable predictions; does this really qualify as "robust predictive power"?

It is true that this might have been an overstatement and we have removed it from the text. Please note that the TCGA analysis corresponding to this paragraph has been re-worked and re-located in the new version of the manuscript.

For Figure 3E and the related text, the authors should elaborate more on this TCGA analysis, for instance, what are the top drug modules for each major tumor type? Are certain approved targeted therapies in one tumor type able to be suggested for other cancers based on this analysis?

We agree with the Reviewer: the original TCGA analysis was shallow and we kindly refer him to similar comments by other Reviewers. In particular, we provide an extensive discussion in response to Reviewer #1 comments (including Figures R2 and

R3). In brief, we have explored possible associations between drug modules and certain tumor types, and we provide a Supplementary Data S6 containing the full result of the drug-tissue association analysis. For example, following a suggestion by Reviewer #1, we found the modules of BRAF inhibitors to be highly enriched in skin cutaneous melanomas (SKCM) (new Figure 4E). Additionally, we performed a similar analysis to relate drug modules to driver mutations identified in the TCGA cohort (Supplementary Data S7).

While, in principle, our drug modules could be used to suggest new therapies, we prefer to down-tone our claims in this regard, given the impossibility to validate predictions due to the limitations of treatment outcome data in the TCGA [22]. We comment and explore this issue in our answers to Reviewer #1. We have re-written significant parts of the text in order to clarify our conclusions about the TCGA analysis.

For better presentation purposes, the authors should consider depicting one or several of the modules in Figure 4A in the same format as illustrated by Figure 2D.

Thanks for the suggestion. We have added a new panel to Figure 4 (Figure R24 here) showing the drug module of lestaurtinib (CEP-701). The network represents the module and illustrates how genes in the module are highly interconnected

Figure R24. Module for the drug lestaurtinib. The module has 40 genes (in red). In addition, two “uncorrelated” proteins are included (in gray) to illustrate the interconnectivity of the module interactome. The protein in the center is highly “connected” to the module, but is not highly expressed in sensitive cell lines; the gray protein in the periphery acts as a “bridge” for the otherwise disconnected CERK1 gene.

The authors should deposit the source code supporting their analysis to Github or a similar repository.

The repository for the module identification analysis is now available at <https://github.com/sbnb-lab-irb-barcelona/GDSC-drug-modules>. The code is documented and straightforward to use. It takes as input a gene expression matrix of a panel of cell lines, and drug sensitivity data across the same panel. A sentence has been written to publicize the repository.

The manuscript has a large number of idiomatic, grammatical, and typographical issues. The authors should proofread carefully and consult readers with strong native English-speaking writing skills as needed for narrative assistance.

Thanks for this comment. The paper has been now revised by two colleagues with native English skills, and we have thoroughly corrected writing issues accordingly.

It is not clear that Figure 1 should be a main figure rather than a supplemental figure. The statements that need to be made about it should be reworked in light of section (1) comments anyway, so we leave this to the authors to reconsider as they move through their re-write.

We have decided to keep Figure 1 in the main text, especially since Reviewer #1 found it of particular interest: “there is a nice presentation of feature analysis that that the authors leverage in their work, such as looking which genes (including a specific analysis on drug targets) are correlated to drug response and how this relates to drug class and to tissue type”. Figure 1 has been reworked and further supported by robustness analysis (Figures S2).

Figure 2A does not appear to be referenced in the text.

It is true that Figure 2A was referenced only in the Methods. We now include a reference to it in the Results section.

The authors should clarify what the vertical axis means in Figure 3B?

The axis in Figure 3B measures the proportion of targets identified at a given distance to the module. We now make this more clear, both in the plot and in the legend.

The text and potentially the figure legend for Figure 3C would benefit from 1-2 sentences on why the diagonal elements are not all identity. Being symmetric this matrix could have been an upper triangular matrix only.

We have improved the figure legend, including a note on why the diagonal elements are not all identity. In addition, following Reviewer's suggestion, we've made the symmetric matrix triangular.

Reviewer #4

The integration of large-scale drug sensitivity screens and genome-wide experiments has revealed molecular determinants of drug response. In particular, transcriptomic signatures of drug sensitivity may guide drug repositioning, prioritize drug combinations and suggest new therapeutic biomarkers. However, the inherent complexity of transcriptomic signatures, with thousands of genes differentially expressed, makes them hard to interpret, giving poor mechanistic insights and hampering translation to the clinics. In this study, the authors show how network biology can help simplify transcriptomic drug signatures, filtering out irrelevant genes, accounting for tissue-specific biases and ultimately yielding functionally-coherent, less noisy drug modules.

The manuscript was well written and the network biology based annotations they provided will be very useful to the scientific community. The authors also found that the network-based method can significantly improve drug-target identification, from 26 to To improve the manuscript, I have following comments.

We thank the Reviewer for his/her positive comments and, in particular, we appreciate that he/she found that our method can improve drug-target identification. Below, we answer to his/her comments point-by-point.

1. Figure 1A, I see the original drug-gene distribution (purple line) is more spread than the permuted distribution (blue). Is there any explanation?

In Figure 1A, the blue distribution represents the null distribution of the correlation analysis, as defined by a random shuffling of the gene expression vs cell line matrix. In contrast, the purple distribution corresponds to the observed correlations. The fact that the latter is a more "spread" distribution can be thus attributed to the existing (non-random) "connection" between drug sensitivity and gene expression. In other words, the width of this distribution demonstrates that gene expression and drug sensitivity correlate more strongly than expected by chance. We have clarified this observation in the Figure legend.

2. Figure 1B. I don't quite understand the figure. Shouldn't Fig. 1B left and 1B right symmetric? Or Maybe the label on X-axis should be "order of genes" and on right "order of drugs". But there are 225 drugs on left and only 70 drugs on the right, please explain.

It is true that Figure 1B was confusing. The left panel shows the “number of correlated genes per drug”, while the right panel shows the “number of correlated drugs per gene”. In the left panel one can read, for example, that there are about 25 drugs (y-axis) with at least 1,250 correlated genes (x-axis). Likewise, in the right panel one can read that about 4,000 genes (y-axis) are correlated to at least 10 drugs (x-axis). We have added this explanation in the figure legend, and modified the x/y-axis labels.

3. Figure 1C. Put the total number of drugs for each drug class. On the "other" class, two drugs have over 2000 correlated genes. It is an unbelievable number. What drugs are these?

Following Reviewer’s suggestion, we have added the number of drugs per drug class in Figure 1C. As for the two modules having a high number of associated genes, these correspond to a single drug ((5Z)-oxozeanol; positively and negatively correlations). This drug was classified by GDSC authors as having a mechanism of action of type “other”. Deeper literature analysis, however, reports (5Z)-oxozeanol as a potent and selective MAPK inhibitor [33]. Interestingly, MAPK inhibitors are the class of drugs that have the largest number of correlated genes (see Figure 1C).

4. There are other efforts on predicting drug-target interactions, such as recent published using deep learning/machine learning approach (PMID: 30347404, PMID: 29917050). The authors should compare the pros and cons of these other methods with their network based methods.

We thank the Reviewer for these references and we kindly refer him/her to our comments to Reviewers #2 and #3 for a comparison of our method to other published approaches using machine learning (e.g. Figure R5 in this document and Supplementary Data S5). The references provided by the Reviewer have been added in the new version of the manuscript as examples of target identification algorithms, together with a discussion about the potential of our models as inputs for machine learning tasks. The “pros and cons” paragraph in the discussion has been extended as well.

5. page 6, the author mentioned they excluded lung cancer, but here they still used NSCLC, confusing.

We apologize for this confusion. We excluded squamous cell lung carcinoma (SCLC) because of identified biases in gene expression (see Methods), but we kept non-squamous cell lung carcinomas (NSCLC). This is now clarified in the text.

6. when calculate the drug-gene correlation, how many data points on average for each drug-gene pair?

On average, each drug was tested on 610 cell lines (drugs tested on less than 400 cell lines were excluded, as recommended elsewhere [26]). The Fisher's z-correction was applied to the Pearson's correlation in order to account for the variable number of cell lines per drug.

References

1. Schadendorf D. Peroxisome proliferator-activating receptors: a new way to treat melanoma? *J Invest Dermatol.* 2009;129:1061–3.
2. Borland MG, Kehres EM, Lee C, Wagner AL, Shannon BE, Albrecht PP, et al. Inhibition of tumorigenesis by peroxisome proliferator-activated receptor (PPAR)-dependent cell cycle blocks in human skin carcinoma cells. *Toxicology.* 2018;404-405:25–32.
3. Ardito CM, Grüner BM, Takeuchi KK, Lubeseder-Martellato C, Teichmann N, Mazur PK, et al. EGF receptor is required for KRAS-induced pancreatic tumorigenesis. *Cancer Cell.* 2012;22:304–17.
4. Tzeng C-WD, Frolov A, Frolova N, Jhala NC, Howard JH, Buchsbaum DJ, et al. Epidermal growth factor receptor (EGFR) is highly conserved in pancreatic cancer. *Surgery.* 2007;141:464–9.
5. Lee Y, Lee J-K, Ahn SH, Lee J, Nam D-H. WNT signaling in glioblastoma and therapeutic opportunities. *Lab Invest.* 2016;96:137–50.
6. Zuccarini M, Giuliani P, Ziberi S, Carluccio M, Iorio PD, Caciagli F, et al. The Role of Wnt Signal in Glioblastoma Development and Progression: A Possible New Pharmacological Target for the Therapy of This Tumor. *Genes [Internet].* 2018;9. Available from: <http://dx.doi.org/10.3390/genes9020105>
7. Rao SK, Edwards J, Joshi AD, Siu I-M, Riggins GJ. A survey of glioblastoma genomic amplifications and deletions. *J Neurooncol.* 2010;96:169–79.
8. Sim H-W, Morgan ER, Mason WP. Contemporary management of high-grade gliomas. *CNS Oncol.* 2018;7:51–65.
9. Kotliarova S, Pastorino S, Kovell LC, Kotliarov Y, Song H, Zhang W, et al. Glycogen synthase kinase-3 inhibition induces glioma cell death through c-MYC, nuclear factor-kappaB, and glucose regulation. *Cancer Res.* 2008;68:6643–51.
10. Higuchi F, Fink AL, Kiyokawa J, Miller JJ, Koerner MVA, Cahill DP, et al. PLK1 Inhibition Targets Myc-Activated Malignant Glioma Cells Irrespective of Mismatch Repair Deficiency-Mediated Acquired Resistance to Temozolomide. *Mol Cancer Ther.* 2018;17:2551–63.
11. Paplomata E, O'Regan R. The PI3K/AKT/mTOR pathway in breast cancer: targets, trials and biomarkers. *Ther Adv Med Oncol.* 2014;6:154–66.
12. Wee ZN, Yatim SMJM, Kohlbauer VK, Feng M, Goh JY, Bao Y, et al. IRAK1 is a therapeutic target that drives breast cancer metastasis and resistance to paclitaxel. *Nat Commun.* 2015;6:8746.
13. Weichert W, Kristiansen G, Winzer K-J, Schmidt M, Gekeler V, Noske A, et al. Polo-like kinase isoforms in breast cancer: expression patterns and prognostic implications. *Virchows Arch.* 2005;446:442–50.
14. Fallah Y, Brundage J, Allegakoen P, Shajahan-Haq AN. MYC-Driven Pathways in Breast Cancer Subtypes. *Biomolecules [Internet].* 2017;7. Available from: <http://dx.doi.org/10.3390/biom7030053>
15. Koren S, Bentires-Alj M. Breast Tumor Heterogeneity: Source of Fitness, Hurdle for Therapy. *Mol Cell.* 2015;60:537–46.

16. Mateo L, Guitart-Pla O, Duran-Frigola M, Aloy P. Exploring the OncoGenomic Landscape of cancer. *Genome Med.* 2018;10:61.
17. Kiessling MK, Curioni-Fontecedro A, Samaras P, Atrott K, Cosin-Roger J, Lang S, et al. Mutant HRAS as novel target for MEK and mTOR inhibitors. *Oncotarget.* 2015;6:42183–96.
18. Van Aelst L, Barr M, Marcus S, Polverino A, Wigler M. Complex formation between RAS and RAF and other protein kinases. *Proceedings of the National Academy of Sciences.* 1993;90:6213–7.
19. Moodie SA, Willumsen BM, Weber MJ, Wolfman A. Complexes of Ras.GTP with Raf-1 and mitogen-activated protein kinase kinase. *Science.* 1993;260:1658–61.
20. Fitzgerald TL, Lertpiriyapong K, Cocco L, Martelli AM, Libra M, Candido S, et al. Roles of EGFR and KRAS and their downstream signaling pathways in pancreatic cancer and pancreatic cancer stem cells. *Adv Biol Regul.* 2015;59:65–81.
21. Goodrich DW, Wang NP, Qian YW, Lee EY, Lee WH. The retinoblastoma gene product regulates progression through the G1 phase of the cell cycle. *Cell.* 1991;67:293–302.
22. Liu J, Lichtenberg T, Hoadley KA, Poisson LM, Lazar AJ, Cherniack AD, et al. An Integrated TCGA Pan-Cancer Clinical Data Resource to Drive High-Quality Survival Outcome Analytics. *Cell.* 2018;173:400–16.e11.
23. Sahu A, Lee JS, Zhang G, Wang Z, Tian T, Moll T, et al. Genome-wide prediction of synthetic rescue mediators of resistance to targeted and immunotherapy [Internet]. 2018. Available from: <http://dx.doi.org/10.1101/284240>
24. Juan-Blanco T, Duran-Frigola M, Aloy P. Rationalizing Drug Response in Cancer Cell Lines. *J Mol Biol.* 2018;430:3016–27.
25. Leiserson MDM, Vandin F, Wu H-T, Dobson JR, Eldridge JV, Thomas JL, et al. Pan-cancer network analysis identifies combinations of rare somatic mutations across pathways and protein complexes. *Nat Genet.* 2015;47:106–14.
26. Rees MG, Seashore-Ludlow B, Cheah JH, Adams DJ, Price EV, Gill S, et al. Correlating chemical sensitivity and basal gene expression reveals mechanism of action. *Nat Chem Biol.* 2016;12:109–16.
27. Iorio F, Knijnenburg TA, Vis DJ, Bignell GR, Menden MP, Schubert M, et al. A Landscape of Pharmacogenomic Interactions in Cancer. *Cell.* 2016;166:740–54.
28. Jaeger S, Duran-Frigola M, Aloy P. Drug sensitivity in cancer cell lines is not tissue-specific. *Mol Cancer.* 2015;14:40.
29. Menche J, Sharma A, Kitsak M, Ghiassian SD, Vidal M, Loscalzo J, et al. Uncovering disease-disease relationships through the incomplete interactome. *Science.* 2015;347:1257601–1257601.
30. Fisher RA. *Statistical Methods for Research Workers.* 1970.
31. Haibe-Kains B, El-Hachem N, Birkbak NJ, Jin AC, Beck AH, Aerts HJWL, et al. Inconsistency in large pharmacogenomic studies. *Nature.* 2013;504:389–93.
32. Ghiassian SD, Menche J, Barabási A-L. A Disease Module Detection (DIAMOND) algorithm derived from a systematic analysis of connectivity patterns of disease proteins in the human interactome. *PLoS Comput Biol.* 2015;11:e1004120.
33. Wu J, Powell F, Larsen NA, Lai Z, Byth KF, Read J, et al. Mechanism and in vitro pharmacology of TAK1 inhibition by (5Z)-7-Oxozeaenol. *ACS Chem Biol.* 2013;8:643–50.